# Atlantic-origin water extension into the Pacific Arctic induced an anomalous biogeochemical event

Shigeto Nishino [1,8] ✉, Jinyoung Jung [2,8], Kyoung-Ho Cho [2], William J. Williams[3], Amane Fujiwara [1], Akihiko Murata [4], Motoyo Itoh[1], Eiji Watanabe [1], Michio Aoyama[4,5], Michiyo Yamamoto-Kawai [6], Takashi Kikuchi[1], Eun Jin Yang [2] & Sung-Ho Kang[7]

The Arctic Ocean is facing dramatic environmental and ecosystem changes. In this context, an international multiship survey project was undertaken in 2020 to obtain current baseline data. During the survey, unusually low dissolved oxygen and acidified water were found in a high-seas fishable area of the western (Pacific-side) Arctic Ocean. Herein, we show that the Beaufort Gyre shrinks to the east of an ocean ridge and forms a front between the water within the gyre and the water from the eastern (Atlantic-side) Arctic. That phenomenon triggers a frontal northward flow along the ocean ridge. This flow likely transports the low oxygen and acidified water toward the high-seas fishable area; similar biogeochemical properties had previously been observed only on the shelf-slope north of the East Siberian Sea.

In recent decades, the Arctic sea ice cover decreased[1]. As the sea ice declines, the response of the upper ocean circulation to winds can be enhanced and the freshwater in the eastern Arctic basins is more likely to shift toward the Beaufort Gyre (BG) region in the western Arctic[2,3] (Fig. 1a), where freshwater accumulation has been observed[4,5]. Bering Strait mooring observations from 1990 to 2019 show increasing northward flow of Pacific water (PW), spring/fall warming, and winter freshening[6]. The anomalous PW influence is called Pacification[7]. The PW transports a large quantity of nutrients; however, freshwater accumulation within the BG deepens the nutrient-rich PW[8] and may decrease phytoplankton production[9]. In contrast, increased eddy activity due to sea ice loss could replenish nutrients and increase phytoplankton production[10]. The freshening of PW inflow in winter may make nutrients available closer to the surface for spring phytoplankton bloom[6].

On the other side, recent changes in the eastern Arctic Ocean are frequently described in terms of Atlantification, which is related to the progression of anomalies carried by Atlantic water (AW) into the eastern Arctic[11] and intensified by the Arctic sea ice decline[12]. Atlantification accompanies the salinification of halocline above the AW layer, weakening of the halocline stratification, shoaling of the AW layer with higher nutrients, and a possible increase in phytoplankton biomass by the vertical nutrient supply[7] and the phytoplankton transport by AW[13]. Thus, the responses of ocean conditions to the sea ice loss are quite different between the western (Pacific) and eastern (Atlantic) Arctic[2,7,14].

The ocean circulation in the Arctic basins plays a vital role in the transports of PW, AW, and Arctic river runoff[15]. The circulation mainly comprises the anticyclonic BG on the Pacific side, cyclonic ocean circulation on the Atlantic side, and Transpolar Drift (TPD), which is roughly the boundary of PW and AW (Fig. 1a). The boundary is climatologically located along the Lomonosov Ridge (LR). However, it shifted eastward to the Mendeleyev Ridge (MR) via the Makarov Basin

[1]Institute of Arctic Climate and Environment Research, Research Institute for Global Change, Japan Agency for Marine-Earth Science and Technology (JAMSTEC), Yokosuka, Japan. [2]Division of Ocean Sciences, Korea Polar Research Institute, Incheon, Republic of Korea. [3]Institute of Ocean Sciences, Fisheries and Oceans Canada, Sidney, BC, Canada. [4]Global Ocean Observation Research Center, Research Institute for Global Change, Japan Agency for Marine-Earth Science and Technology (JAMSTEC), Yokosuka, Japan. [5]Center for Research in Isotopes and Environmental Dynamics, University of Tsukuba, Tsukuba, Japan. [6]Department of Ocean Sciences, Tokyo University of Marine Science and Technology, Tokyo, Japan. [7]Korea Polar Research Institute, Incheon, Republic of Korea. [8]These authors contributed equally: Shigeto Nishino, Jinyoung Jung. ✉ e-mail: nishinos@jamstec.go.jp

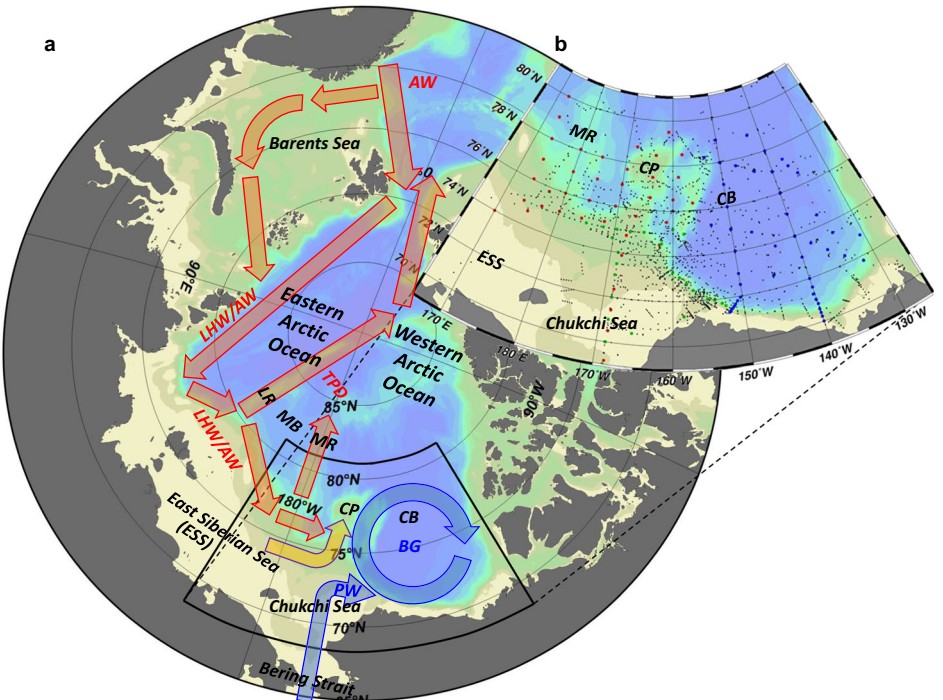

**Fig. 1 | Schematic of the Arctic Ocean circulation and the study area with hydrographic stations. a, b** Maps of the Arctic Ocean and the study area. In **a**, yellow, blue, and red arrows represent flows from the shelf-slope at the north of the East Siberian Sea (ESS), and from the Pacific and Atlantic oceans in 2017–2020. Ocean circulation and water masses are abbreviated as follows: Beaufort Gyre (BG), Transpolar Drift (TPD), Pacific Water (PW), Lower Halocline Water (LHW), and Atlantic Water (AW). Geographical locations are abbreviated as follows: Canada

Basin (CB), Chukchi Plateau (CP), Mendeleyev Ridge (MR), Makarov Basin (MB), and Lomonosov Ridge (LR). In **b** red, green, and blue dots denote the hydrographic stations conducted by the Research Vessel (R/V) Araon (Korea), R/V Mirai (Japan), and Canadian Coast Guard Ship Louis S. St-Laurent (Canada), under the 2020 Synoptic Arctic Survey project. Black dots indicate other hydrographic stations between 2002 and 2019 listed in Supplementary Table 1.

(MB) with enlarged cyclonic ocean circulation and shrinkage of the BG in the Canada Basin (CB) when Arctic atmospheric circulation was characterized by a strong cyclonic mode, i.e., Arctic Oscillation (AO)[16] was in a positive phase (AO+) with its significant signature from the end of the 1980s to the mid-1990s[2,17]. Subsequently, the AO indicated a negative phase (AO−) on average, and it again changed to AO+ after the mid-2010s. Accordingly, the abovementioned ocean circulation shift occurred with an eastward extension of Atlantic-origin water (Lower Halocline Water (LHW) and AW) along the shelf-slope of the East Siberian Sea (ESS) and Chukchi Sea[18]. The Atlantic-origin water intrusion in 2017 pushed up the nutrient-rich PW and ESS water (ESSW), resulting in anomalously high surface phytoplankton blooms[19]. The ESSW transport up to the Chukchi Sea (and further into the CB) is consistent with the previous modeling study, suggesting that the Eurasian runoff leaves the Siberian shelf at more eastern locations under AO+[15]. However, due to a lack of observed data, we have little knowledge about the changes in ocean environments, especially biogeochemical aspects, in the region toward the north of the ESS and the Chukchi Sea including the Chukchi Plateau (CP), an ocean ridge north of the Chukchi Sea. In this region, PW, AW, and ESSW complicatedly intersect (hereafter referred to as the intersectional water region).

In 2020, a multiship sampling campaign covering a broad research area in the western Arctic Ocean (Fig. 1b), including the intersectional water region, was conducted under the Synoptic Arctic Survey (SAS)[20]. During the survey, unusually low dissolved oxygen and acidified water were observed on the CP for the first time. Herein, we describe these biogeochemically abnormal conditions and propose a mechanism driving these changes (Fig. 2a–c) by combining data obtained by the Research Vessel (R/V) Araon (Korea), R/V Mirai (Japan), and Canadian Coast Guard Ship (CCGS) Louis S. St-Laurent

(Canada) in the SAS project. Describing the presence and drivers of these biogeochemical anomalies is a vital preliminary step for Arctic ecosystem assessments[21,22]; we can assess ecosystem responses only when we know the hydrographic changes underpinning them. This information will ultimately help to develop policies facilitating effective management, e.g., for potential fisheries in the future ice-free Arctic Ocean[23].

## Results

### Anomalous water found on the Chukchi Plateau

In 2020, we found low dissolved oxygen and thus, low oxygen saturation (Fig. 3a) in water on the southern portion of the CP (-75° N, 168° W). The low oxygen saturation water also corresponded to acidified water, which was corrosive to aragonite (i.e., saturation state $\Omega_{arg} < 1$; Fig. 3b). The time series of oxygen saturation in a water column (100–300 m) surrounding the CP (172–164° W and 73.5–76.5° N), which is obtained through ship-based observations during the summer and autumn of 2002–2020 (Supplementary Table 1), shows that the oxygen saturation was typically 70–80% (Fig. 3c). In contrast, water with oxygen saturation levels <50% (i.e., oxygen concentrations below 180 μmol kg$^{-1}$) was found in 2020. Such water had never been observed before, at least in the last two decades. Here, we term this water anomalously low oxygen saturation (dissolved oxygen) water or anomalous water.

The anomalously low oxygen saturation water exhibited extremely low $\Omega_{arg}$ values (<0.6) on the CP (Fig. 3). This suggests a high quantity of organic matter decomposition that used dissolved oxygen and produced $CO_2$, resulting in significant decreases in oxygen saturation and $\Omega_{arg}$, respectively. However, there is no evidence of high-level suspended sediments and particles that might promote local microbial oxygen consumption and $CO_2$ production, unlike the

conditions observed in the ESS where much of terrestrial/marine organic matter is deposited on the bottom[24,25].

Historical data[26,27] indicate that the water with oxygen saturation as low as the anomalous water on the CP occupied the bottom along the shelf-slope north of ESS between 170° E and 170° W, hereafter referred to as ESS shelf-slope (Fig. 4a). In shallow shelves, currents and storm events replenish dissolved oxygen and diffuse $CO_2$ in the bottom water[25]. On the shelf-slope west of 170° E, AW would penetrate from the west along the slope. PW occupied east of 170° W. The AW and PW were likely well-ventilated in winter[28], so dissolved oxygen concentrations could be high. Consequently, the zone of the lowest oxygen saturation along the ESS shelf-slope is considered a region where AW and PW influences were minimal. Thus, it is recognized as a shadow (less ventilated) zone of the AW and PW inflows. The shadow zone is consistent with the lowest concentrations of an artificial inert gas, sulfur hexafluoride ($SF_6$), in the ESS shelf-slope[29]. The bottom salinity of the shadow zone (Fig. 4b) was similar to that of the anomalous water on the CP. In the cold Arctic seas, salinity mainly determines the water density. It is used to identify water masses that should move along the same density surfaces. In addition, biogeochemical characteristics of the lowest oxygen saturation water along the ESS shelf-slope were similar to those of the anomalous water on the CP (Supplementary Discussion 1 and Supplementary Figs. 1 and 2). Thus, the shadow zone of the ESS shelf-slope is the likely source of the anomalous water on the CP.

Historical data indicate that the lowest oxygen saturation levels along the ESS shelf-slope were observed in winter and summer. Thus, the shadow zone would be less ventilated throughout the year. Anomalously low oxygen saturation water on the CP was found in October 2020. Only with this observation it is not easy to describe the seasonality of the anomalous water. This will be examined using our numerical model below.

## Changes in PW and AW circulation

We hypothesize that a change in ocean circulation in the intersectional water region north of ESS and Chukchi Sea caused the anomalous event on the CP, such as the appearance of low oxygen saturation water, which was likely derived from the ESS shelf-slope. The Arctic upper ocean circulation variation is mainly associated with the Beaufort High (BH) variability and AO[2]. The BH induces a negative curl of winds that converges the surface freshwater within the BG, and the spatial distribution of freshwater determines the extent of the BG. On the other hand, when AO is positive, the BG shrinks to a smaller area in the western Arctic, and the cyclonic ocean circulation in the eastern Arctic expands toward the western Arctic[30]. The wind curl over the CB was unprecedentedly negative in the second half of the 2000s and 2010s, and AO changed its phase from negative to positive in mid-2010s[2]. During the 2000s and 2010s, although the BG extent evaluated via numerical simulations[31] and satellite data[19,32] generally increased until 2016 (the increase in sea surface height of the gyre had slowed over the 2010s[33]), the gyre shrunk in 2017 and the cyclonic ocean circulation expanded concurrently under AO+[19,31].

Here, the circulation pattern and oxygen saturation distribution in 2017–2020 are compared with those in 2008–2016, selected as a period when the BG extent was larger than that in 2017–2020. These circulation patterns and oxygen saturation distributions are also compared with those obtained from historical data (temperature and salinity data[27] and oxygen data[26]) that were mainly obtained during the 1950s to 1980s, when the wind curl over the CB had a positive anomaly on average and the AO was generally in a negative phase. In the Discussion section, we will further mention the hydrographic condition and oxygen saturation distribution in the mid-1990s during a highly positive phase of AO. The circulation pattern in each period is calculated as the dynamic height at a depth of 100 m, which is a suitable depth to examine the PW and AW circulation as used in previous studies[34] and described below. Because the observation points in each period are not spatially uniform, comparisons between hydrographic conditions with different data spacings may lead to a distortional feature. Thus, the observational data were interpolated to gridded points in each period for evaluating dynamic height (Fig. 5a–c), temperature and salinity related to the PW and AW circulation (Fig. 5d–f), and oxygen saturation distributions (Fig. 5g–i). The interpolation uncertainties inherent to each parameter were also estimated (Supplementary Discussion 2 and Supplementary Figs. 3 and 4).

In 2017–2020 (Fig. 5a), an anticyclonic circulation, the BG, occupied east of the CP (-168° W), and northward flows were predominant

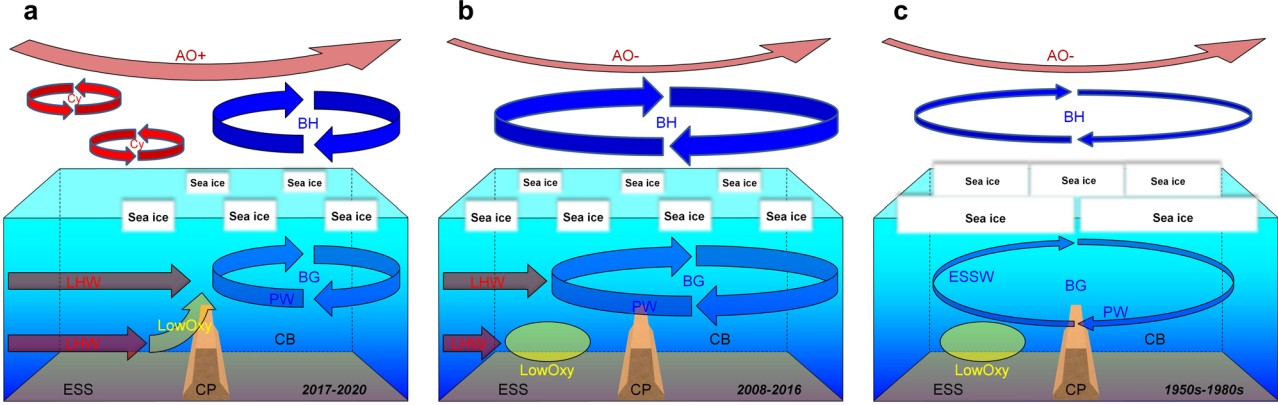

**Fig. 2 | Hypothesized mechanism of distribution change in low oxygen water. a** 2017–2020: the atmospheric condition was characterized by a positive Arctic Oscillation phase (AO+; a thick red arrow at the top) and a small extent of the Beaufort High (BH; upper blue arrows) with a strongly negative wind curl (thick arrows of the BH). The intrusion of North Atlantic cyclones (Cys; small red arrows) could contribute to AO+. This atmospheric condition caused a shrink of the Beaufort Gyre (BG; lower blue arrows), which transported Pacific water (PW) to the east of the Chukchi Plateau (CP) in the Canada Basin (CB). Lower Halocline Water (LHW) from the eastern Arctic penetrated near the CP (lower red arrows) under AO+. A northward flow along the CP (a yellow allow) caused by the PW/LHW front carried low oxygen (LowOxy) water from the shelf-slope at the north of the East Siberian Sea (ESS) to the CP. **b** 2008–2016: a negative Arctic Oscillation phase (AO−; a thin red arrow at the top) and a large extent of the BH caused BG enlargement to the west of the CP with PW extension. LHW could not penetrate the CP under AO−. LowOxy water formed on the shelf-slope at the north of the ESS (yellow ellipse) between the oxygen-rich PW and LHW influences. **c** The 1950s–1980s: AO− and a large extent of the BH with a weakly negative wind curl (thin arrows of the BH). Sea ice cover inhibited the input of wind curl to the ocean. The BG was wide, but its circulation was weak. The BG transported the PW to the west of the CP and the ESS water (ESSW) to the north of the ESS. The penetration of LHW was minimal as the AO was negative and Atlantification had not yet occurred. Thus, LowOxy water occupied the shelf-slope at the north of the ESS.

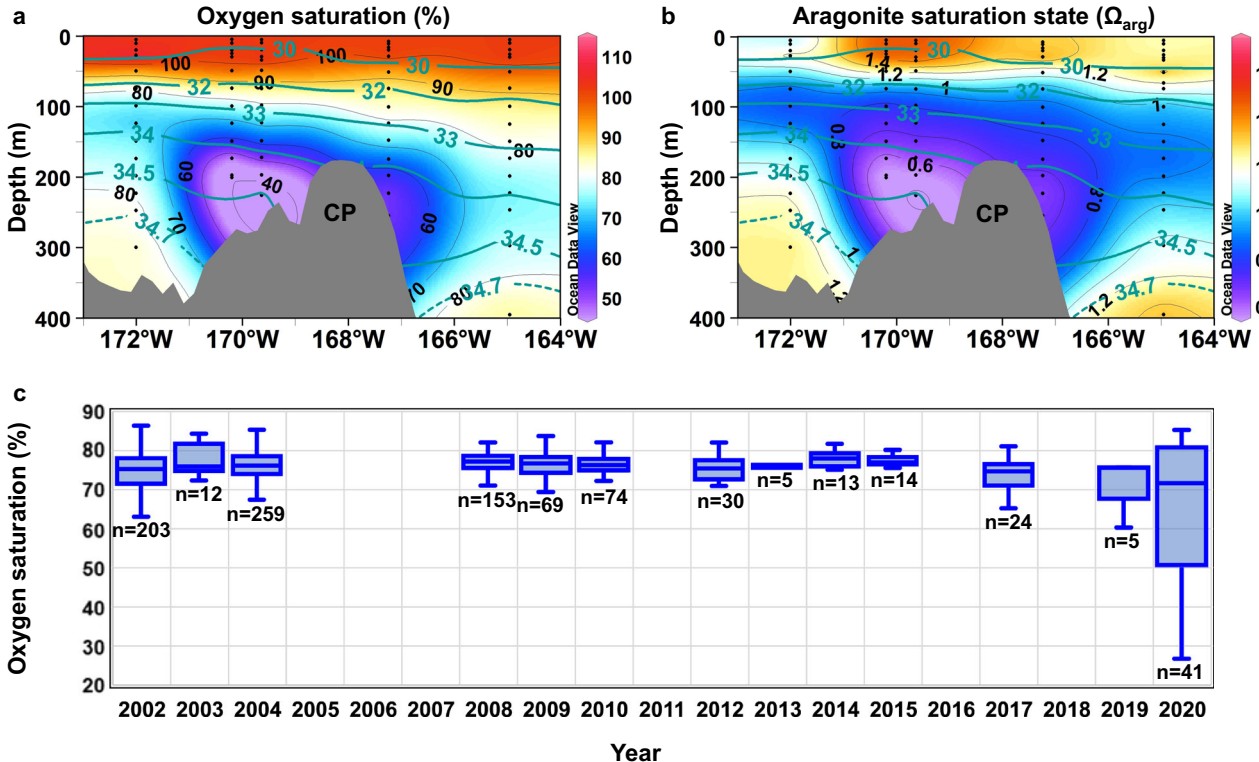

**Fig. 3 | Water on the Chukchi Plateau in 2020 is characterized by anomalously low oxygen saturation and is corrosive to aragonite. a, b** Vertical sections of oxygen saturation and aragonite saturation state, $\Omega_{arg}$, respectively, along 75° N across the Chukchi Plateau (CP; ~168° W) measured during the Research Vessel Mirai cruise in 2020. Salinity contours are overlain, and black dots indicate the data points comprising each vertical section. **c** Box and whisker plot showing yearly variations in oxygen saturation. Boxes indicate the lower and upper quartiles. The horizontal line in each box represents the median oxygen saturation. Vertical lines extending from each box represent the minimum and maximum values recorded for that year. The number of samples for each year is shown below the minimum line. The samples were collected between the depth range of 100 and 300 m in the water column around the CP in an area from 172 to 164° W and from 73.5 to 76.5° N. They were obtained through ship-based observations as listed in Supplementary Table 1. Source data are provided as a Source Data file.

west of the plateau. The CP could be a western boundary of the BG[35]. The northward flow west of the CP might be a branch of cyclonic ocean circulation extended from the Eastern Arctic. The BG shrunk toward the southeast compared to 2008–2016 (Fig. 5b, Supplementary Fig. 4a). In 2008–2016 (Fig. 5b), the westward flow of the southern part of the BG overshot the southern CP, and the return flow passed across the northern CP. The shrinkage of the BG in 2017–2020 is likely associated with the collapse of the BH in 2017[36]. In this period, the area of the strong negative wind curl was confined to the southeastern CB[2]. In the 1950s–1980s (Fig. 5c), the BG circulation was weak because the wind curl over the CB was not strongly negative in this period[2]. A heavy and less mobile sea ice condition in this period might also inhibit the input of wind curl to the ocean[37]. Because the AO was a negative phase in this period, the cyclonic ocean circulation would not expand to the study area.

The circulation patterns are associated with temperature and salinity distributions along 75–76° N across the CP (Fig. 5d–f). The BG transports cold PW that has entered the Arctic Ocean through the Bering Strait in winter and is characterized by a temperature minimum around salinity = ~33[38]. Cold ESSW (salinity = 32–33) occupies the west of the cold PW[39]. Layers below the cold waters are occupied by LHW derived from the eastern Arctic (salinity = 34–34.5)[38,40] and warm AW (temperature > 0 °C, salinity > 34.5)[41], both of which are transported by the cyclonic ocean circulation. In 2017–2020 (Fig. 5d), a strong front between the cold PW and eastern Arctic-derived LHW was found at depths of 100–150 m just west of the CP with a shoaling of the AW. This front caused a significant northward flow there (Figs. 2a and 5a). Such a frontal structure was not found in 2008–2016 (Fig. 5e) because the cold PW penetrated further west of the CP with the extension of the BG

(Figs. 2b and 5b). In 2017–2020, shallow levels of LHW and AW at the west of the CP suggest enlarged intrusions of LHW and AW. The enlarged LHW and AW intrusions were presumably associated with the intensification of AW supply induced by the sea ice decline (Atlantification)[12] and AO+, which enhanced the cyclonic ocean circulation, in the second half of the 2010s. With the strengthening and expansion of the cyclonic ocean circulation and the shrinkage of the anticyclonic BG circulation, the LHW and AW could further flow along isobaths and spread into the CB[42,43]. However, the spreading of these water masses, especially the LHW, was likely hampered by the BG, whose western boundary appeared over the CP when the gyre shrunk. Thus, there was a strong front between the PW and LHW in 2017–2020. In the 1950s–1980s (Fig. 5f), the cold PW with salinity = ~33 extended to the west of the CP. This cold PW would be carried by westward flows of the BG circulation, which was weak but widely expanded under AO− (Figs. 2c and 5c). The cold water with salinity = 32–33 found on the western side of the section would be derived from the ESS by northward flows of the BG circulation. The intrusions of LHW and AW were likely small because of AO− and the condition before Atlantification occurred, as shown in a deep AW layer and low temperature of the AW.

**Oxygen saturation distribution determined by ocean circulation**

The concurrent expansion of the cyclonic ocean circulation and shrinkage of the BG would determine the fate of the water originally located on the ESS shelf-slope. We examined the oxygen saturation distributions on isopycnal (i.e., the same potential density) surfaces of $\sigma_\theta = 27.7$ (S ~ 34.5) in 2017–2020 and 2008–2016 (Fig. 5g, h) and $\sigma_\theta = 26.5$ (S ~ 33) in the 1950s–1980s (Fig. 5i). Each isopycnal surface is located in a layer including the lowest oxygen saturation water in each

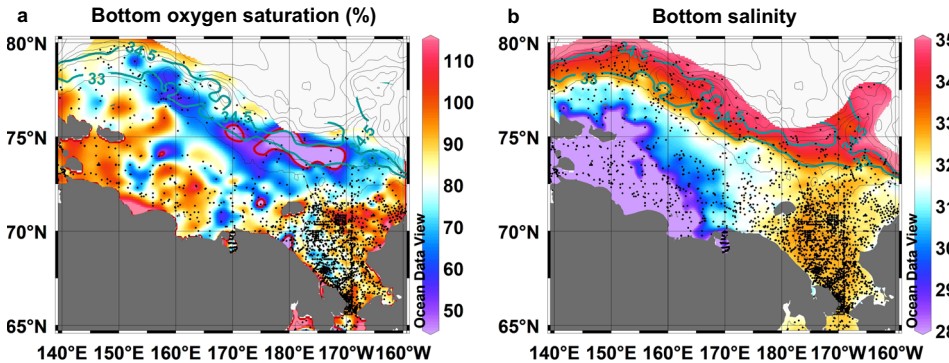

**Fig. 4 | Shadow zone of Pacific and Atlantic water inflows is characterized by the lowest oxygen saturation along the shelf-slope north of the East Siberian Sea.** **a** Oxygen saturation (%) and **b** salinity at the bottom (depth <500 m) of the shelf and shelf-slope regions of the East Siberian and Chukchi seas. Black dots indicate data points. In **a**, a region in which water with the lowest oxygen saturation (<50%) was found along the shelf-slope north of the East Siberian Sea is enclosed by a red contour. In **a** and **b**, salinity contours of 33 and 34.5 are shown as green lines. Oxygen data were obtained from the Hydrochemical Atlas of the Arctic Ocean[26]. Salinity data were obtained from the Environmental Working Group Joint US–Russian Atlas of the Arctic Ocean[27]. Source data are provided as a Source Data file.

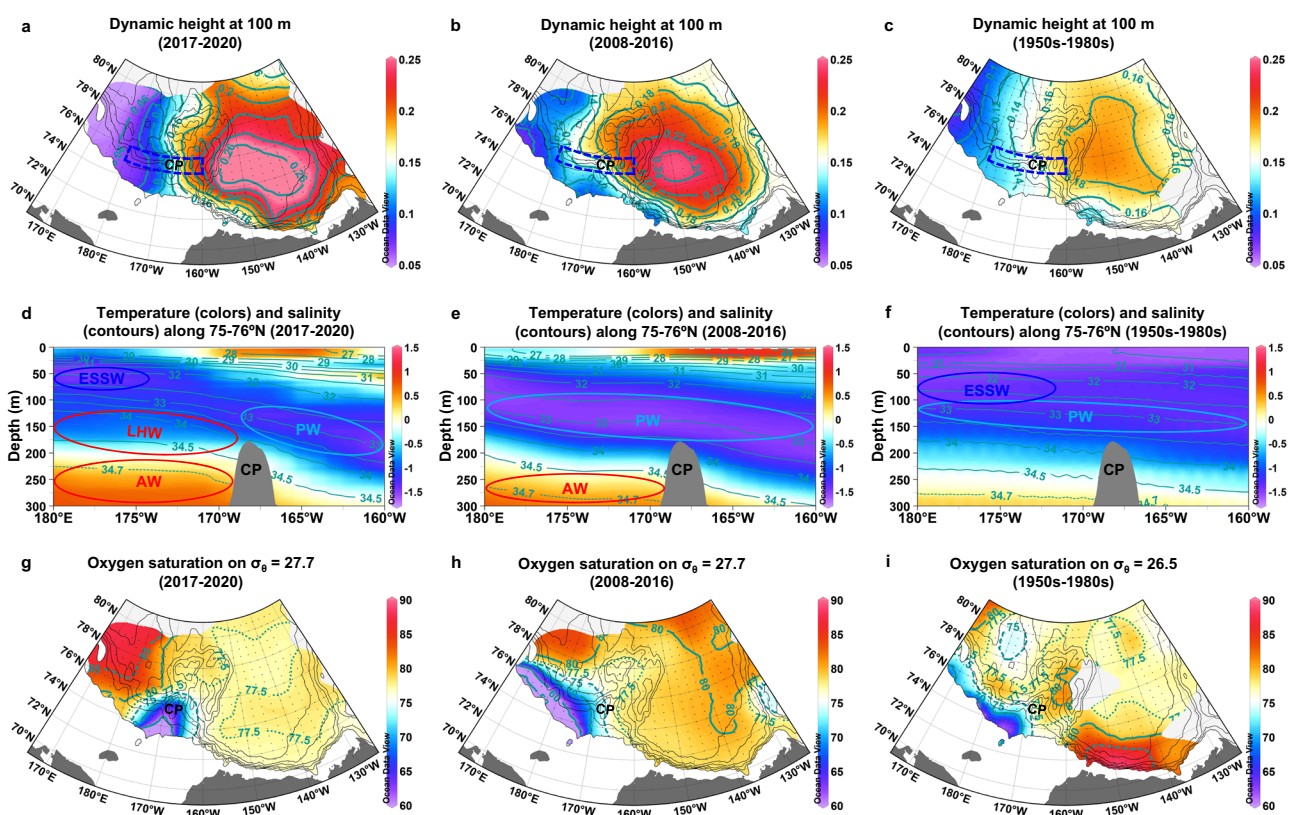

**Fig. 5 | Changes in ocean circulation and distributions of temperature, salinity, and oxygen saturation in a region influenced by Pacific and Atlantic waters.** Dynamic height (dyn m) at 100 m relative to 250 m in **a** 2017–2020, **b** 2008–2016, and **c** the 1950s–1980s. Black dots represent 0.5° × 2.5° latitude–longitude gridded points where interpolation values can be calculated using data from the cruises listed in Supplementary Table 1 and historical data (see Methods). Chukchi Plateau is abbreviated as CP. **d**–**f** Vertical sections of temperature (°C; colors) and salinity (contours) along a band of 75–76° N (blue dashed square in **a**–**c**) with a 1.0° longitude grid in **d** 2017–2020, **e** 2008–2016, and **f** the 1950s–1980s. **g**–**i** Oxygen saturation (%) along isopycnal surfaces of **g** $\sigma_\theta$ = 27.7 in 2017–2020, **h** $\sigma_\theta$ = 27.7 in 2008–2016, and **i** $\sigma_\theta$ = 26.5 in the 1950s–1980s. Black dots indicate 0.5° × 2.5° latitude–longitude gridded points where interpolation values can be calculated. Source data are provided as a Source Data file.

case (Fig. 4a, Supplementary Figs. 1a and 2a). The isopycnal surface of $\sigma_\theta$ = 27.7 (S ~ 34.5) in the 1950s–1980s was slightly deeper than the level of the lowest oxygen saturation water on the ESS shelf slope (Fig. 4a). Thus, we selected a shallower isopycnal surface ($\sigma_\theta$ = 26.5, S ~ 33).

The oxygen saturation distributions in 2008–2016 (Fig. 5h) and the 1950s–1980s (Fig. 5i) show that the area with the lowest oxygen saturation on the isopycnal surface appeared along the ESS shelf-slope.

This area is thought to be a shadow zone. However, in 2017–2020 the area with the lowest oxygen saturation moved to the CP (Fig. 5g). The low oxygen saturation around the CP was the most pronounced in 2020, i.e., the anomalously low oxygen saturation water was found there (Fig. 3a). It further extended toward the north with increasing oxygen saturation. The migration of the area with the lowest oxygen saturation suggests that the lowest oxygen saturation water in the

shadow zone was washed by an eastward penetration of LHW (i.e., higher oxygen saturation water) along the ESS shelf-slope. Thus, the shadow zone there was no longer maintained. The washed-out lowest oxygen saturation water was likely transported to the north by the strong frontal flow between the PW and LHW over the CP. Because the biogeochemical characteristics between the lowest oxygen saturation water along the ESS shelf-slope and the anomalously low oxygen saturation water on the CP were almost the same, the lowest oxygen saturation water was not considerably modified by mixing with the surrounding water during its migration to the CP. In contrast, the water was substantially modified after leaving the CP along the frontal northward flow. Assuming horizontal and vertical diffusivities of $10^2$ [44] and $10^{-6}$ [45] $m^2 s^{-1}$, respectively, and estimating the horizontal and vertical gradients of oxygen concentration from the observed data over the CP in 2020, we found that oxygen concentrations increased along the frontal northward flow primarily through horizontal diffusion (O $(10^{-6})$ $\mu mol \ kg^{-1} \ s^{-1}$), which was an order of magnitude larger than vertical diffusion (O $(10^{-7})$ $\mu mol \ kg^{-1} \ s^{-1}$).

The CB's oxygen saturation was lower in 2017–2020 than in 2008–2016 (Fig. 5g, h). This might be partly explained by the anomalously low oxygen saturation water spreading from the CP into the CB. However, the surrounding water would rapidly dilute the anomalous water through horizontal diffusion north of the CP. Therefore, it could not have contributed to the decreasing oxygen saturation throughout the CB. In 2008–2016, a source of high oxygen saturation seemed to develop in the northeastern CB. A tongue of high oxygen saturation extended southward from the source, which was probably transported by a southward flow of the eastern part of the BG. The source and the tongue of high oxygen saturation were not observed in 2017–2020. Although water with a high oxygen saturation is thought to originate in the eastern Arctic, further study is required to explain why the water was not delivered to the northeastern CB in 2017–2020. The lower oxygen saturation levels in the CB in 2017–2020 than in 2008–2016 were observed in the whole LHW (salinity = 34–34.5) layer (Supplementary Fig. 5a, b). In addition to the absence of high oxygen saturation source in the northeastern CB in 2017–2020, vertical mixing between the LHW layer and the above layer characterized by a vertical minimum of oxygen saturation might be another reason to reduce the oxygen saturation in the LHW layer (Supplementary Discussion 3 and Supplementary Fig. 5). Contrary to the CB, the MB indicated that the oxygen saturation was higher in 2017–2020 (Fig. 5g) than in 2008–2016 (Fig. 5h). This shift would result from the penetration of LHW with a high oxygen saturation to the MB by the expanded cyclonic ocean circulation in 2017–2020 under AO+.

### Modeled transport of the lowest oxygen saturation water

A tracer experiment was conducted using a physical sea ice–ocean general circulation model to evaluate whether the lowest oxygen saturation water on the ESS shelf-slope was transported to the CP (see Methods). A virtual passive tracer was continuously provided on the ESS shelf-slope during two periods, namely, 2013–2016 and 2017–2020 (4 years each), to examine differences in the distribution of low oxygen saturation water between the two periods. A snapshot of the model output in December 2017 (Fig. 6a) shows that the lowest oxygen saturation water on the ESS shelf-slope, where the tracer concentration value was fixed to 1, is transported to the CP by a northward flow. Associated with this transport, the appearance of low oxygen saturation water passing through the CP has become more pronounced since 2017 (Fig. 6b), which is consistent with the year when the observed BG shrunk in 2017. Our numerical model also indicates the shrinking of the BG in 2017–2020 (Supplementary Fig. 6a) compared to 2013–2016 (Supplementary Fig. 6b). The appearance of low oxygen saturation water is not continuous throughout 2017 and beyond (Fig. 6b); instead, it is intermittent on a timescale of several months. The tracer concentration of the water

seems to represent a seasonality with its increase in winter when winds are generally strong [46,47]. Although we observed anomalously low oxygen saturation water on the CP only in October 2020, it might also have been present between 2017 and 2019 when observations were not conducted.

From 2017 to 2020, the mean velocity field in the cross-section from the top of the CP to its western slope (Fig. 6c) shows an overall northward flow, carrying the lowest oxygen saturation water from the ESS shelf-slope. In contrast, southward flows are predominant from 2013 to 2016 (Fig. 6d), and the lowest oxygen saturation water on the ESS shelf-slope hardly spreads to the CP. This southward flow is part of a meander around the southern CP generated when the plateau diverts a westward flow of the BG (Supplementary Fig. 6b). The northward flow in 2017–2020 is likely related to a frontal salinity structure, such as the salinity contours of 34.5 and 34.7, deepening toward the CP (Fig. 6c) as observed in 2017–2020 (Fig. 5d). These isohaline surfaces are shallower in 2017–2020 (Fig. 6c) than in 2013–2016 (Fig. 6d), suggesting an enhanced penetration of AW. In the observed data of 2017–2020 (Fig. 5d), LHW (salinity = 34–34.5) was also shallower than in previous years; however, the simulated isohaline surface of 34, i.e., the upper boundary of the LHW, is almost at the same level in 2013–2016 and 2017–2020. Thus, the model does not accurately reproduce the LHW penetration. The simulated penetration of LHW from the eastern Arctic might be underestimated because of the closed lateral boundary condition in the North Atlantic and insufficient vertical resolution. In addition, the lack of interannual variability and multi-decadal trends in riverine freshwater discharges might produce an unexpected model bias of stratification in the eastern Arctic (isohaline-surface structure of the LHW). Further approaches are needed to reproduce the penetration of properly stratified LHW to the CP, one of the key processes to form the northward flow that transports the lowest oxygen saturation water from the ESS shelf-slope.

## Discussion

The present study suggests that the concurrent shrinkage of the BG and the strengthening of the cyclonic ocean circulation facilitate the transport of the low oxygen saturation water, which had previously only ever appeared in a shadow zone on the ESS shelf-slope, to the CP. The Arctic sea ice decline likely intensifies both the BG and cyclonic ocean circulation [2]. The accompanying shift of freshwater from the cyclonic ocean circulation region to the BG region develops Atlantification in the eastern Arctic, weakening the LHW stratification and the AW uplift. If the cyclonic ocean circulation carried such signals of Atlantification and encountered the BG occupying the east of the CP, a frontal northward flow would be produced along the plateau. Thus, the sea ice decline could increase the low oxygen saturation water transport from the ESS shelf-slope to the CP.

Furthermore, the decreasing sea ice in the eastern Arctic and the resultant intrusion of North Atlantic cyclones into the western Arctic caused the BH collapse, which was first observed in 2017 [36]. Increased cyclones and an accompanying weakening of the BH (decrease in sea level pressure) were also observed after 2017 [48]. The increased cyclones facilitated AO+. They thus enlarged the cyclonic ocean circulation toward the BG region. This effect could also contribute to the northward transport of the low oxygen saturation water.

The previous hydrography observations indicated that the strengthening of the cyclonic ocean circulation and the resultant shift of the boundary between the PW and AW toward the MR occurred in the mid-1990s following the strongly positive AO [49–51]. Biogeochemical data were limited in this period, but oxygen measurements from the Arctic Ocean Section in 1994 showed that water with low oxygen saturation levels of <50% (oxygen concentrations less than 180 $\mu mol \ kg^{-1}$) was found on the Chukchi Sea shelf-slope [52]. Although the low oxygen saturation water seemed to spread toward the north, the oxygen saturation near the CP (~75° N) was not as low as that found in

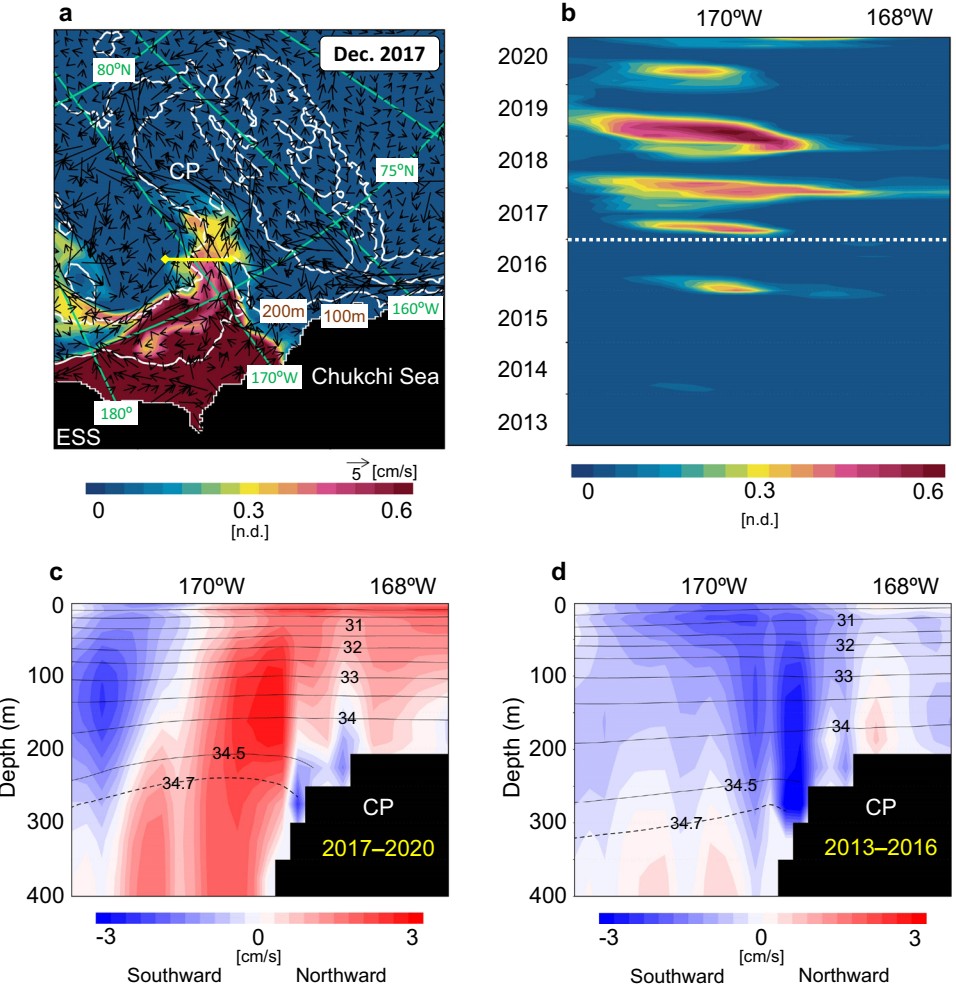

**Fig. 6 | Simulated ocean velocity and tracer distributions to verify the spreading of the lowest oxygen saturation water from the shelf-slope north of the East Siberian Sea. a** Simulated ocean velocity and tracer concentration averaged 100–200 m in December 2017. The simulated tracer distribution represents the spreading of the lowest oxygen saturation water from the shelf-slope north of the East Siberian Sea (ESS), where the tracer concentration value is fixed to 1 (see Methods). The yellow line from the top of the Chukchi Plateau (CP) to its western slope corresponds to the horizontal axis range in **b**–**d**. **b** Time series of tracer concentration of water passing through the yellow line in **a** averaged 100–200 m. **c**, **d** Vertical sections of northward velocity (colors) and salinity (contours) for the periods of 2017–2020 and 2013–2016, respectively, along the yellow line in **a**. Source data are provided as a Source Data file.

2020 (Supplementary Fig. 7). In the mid-1990s, the wind curl over the CB was not strongly negative, and thus the BG was weak. Under weak BG and strongly positive AO, the AW could further penetrate the CB[15]. Therefore, the low oxygen saturation water was probably carried toward the CB (eastward) rather than the CP (northward). Shrinkage of the BG with strong northward flow along the CP is needed to transport the low oxygen saturation water toward the CP.

The lowest oxygen saturation water on the shelf-slope was found between 170° E and 170° W with a 1-degree width and a 50-m thickness (Fig. 4a, Supplementary Fig. 2a); thus, its volume is estimated to be $3.2 \times 10^{12}$ m³. Assuming that, in 2020, the lowest oxygen saturation water occupied 168–170° W with a 100-m thickness on the CP (Fig. 3a) and the northward velocity of the frontal flow had a mean value of ~$2 \times 10^{-2}$ m s⁻¹ (Fig. 6c), the annual volume of the lowest oxygen saturation water that passed through the CP is estimated to be $3.6 \times 10^{12}$ m³. Therefore, it takes approximately one year for the lowest oxygen saturation water volume on the shelf-slope to be removed by the frontal northward flow along the CP. This northward transport of the lowest oxygen saturation water probably started in 2017, as shown in our model (Fig. 6b). Considering the timescale of ~1 year for removing the lowest oxygen saturation water from the shelf-slope, water with a higher oxygen saturation could have occupied the shelf-slope for a certain period (see below) after the end of 2017. In this period, the water with a higher oxygen saturation would be transported to the CP, and therefore, the lowest oxygen saturation water might no longer be observed on the CP.

The oxygen uptake rate in the bottom sediments of outer ESS in 2014 was estimated at 7.2 mmol m⁻² day⁻¹, an order of magnitude greater than 20 years earlier[53]. Suppose the LHW with high oxygen (300 µmol kg⁻¹) and a 50-m thickness has contacted the bottom. Under the oxygen uptake rate of 2014 (or 20 years earlier), it would take 2 (or ~20) years to reduce the oxygen of the water column to 180 µmol kg⁻¹, equivalent to a level of anomalously low oxygen saturation water. Therefore, once the LHW supply from the eastern Arctic to the ESS shelf-slope is ceased for more than two years, low oxygen saturation water will again appear on the shelf-slope under the oxygen uptake rate of 2014. The oxygen uptake rate might increase due to increased organic matter inputs from rivers, coastal erosion, and biological production, resulting in the quicker and wider formation of the lowest oxygen saturation water. Organic matters from rivers and coastal erosion are mostly terrigenous and accumulate in near-shore regions in the western ESS, locally producing low oxygen and acidified water[54]. In the eastern ESS and on the shelf-slope, biological production due to sea ice loss is expected to

increase, and the increased production will amplify organic matter deposition and decay[53,55].

Moreover, the appearance of the frontal northward flow along the CP might increase with the sea ice reduction as described above. Consequently, the anomalous low oxygen and acidification event on the CP could occur more frequently. It may impact the marine ecosystem around the plateau.

The CP is part of a shallow water area termed the Chukchi Borderland. It is considered that areas shallower than 2000 m are fishable in future ice-free high seas of the central Arctic Ocean[21,23]. The Chukchi Borderland is a potential spawning area for Arctic cod, and polar cod may distribute south of the Chukchi Borderland[22]. Both are important cod species in Arctic marine food webs and are of commercial interest. Therefore, when introducing appropriate ecosystem-based management under the Central Arctic Ocean Fisheries Agreement ([www.fao.org/faolex/results/details/en/c/LEX-FAOC199323](www.fao.org/faolex/results/details/en/c/LEX-FAOC199323)) that has recently entered into force, it will become essential to monitor the marine environment and ecosystem in this region, especially across the CP where anomalous events of low oxygen water and acidification are expected to increase in the future. At present, the lowest oxygen concentration found on the CP in 2020 was ~100 $\mu$mol kg$^{-1}$ (~30% in oxygen saturation), which is higher than the general threshold of hypoxia (60 $\mu$mol kg$^{-1}$) capable of causing physiological stresses or mortality events for marine organisms, but in a range (as high as ~130 $\mu$mol kg$^{-1}$) affecting the growth and behavior of some fish species[56]. Cold-water fishes tend to have higher hypoxia thresholds[57]. The Greenland cod, which distributes from Alaska to the western coast of Greenland, has a hypoxia threshold of ~40% in oxygen saturation at 1 °C, and it would be higher oxygen saturation levels with ocean warming[58]. The same species in the Bering and Chukchi seas, i.e., the Pacific cod, is expected to expand northward if warming conditions continue, based on observational[59] and model[60] studies. However, such expansion might be inhibited by the anomalously low oxygen saturation water, which has already appeared on the CP and is assumed to be enlarged in the future. Future projections under ocean warming and oxygen loss suggest that extinction risks for marine animals are highest (lowest) in polar (tropical) species[61]. This latitudinal extinction pattern is consistent with fossil records of the end-Permian extinction[62,63].

Ocean acidification is one of the prioritized indicators associated with the Central Arctic Ocean Fisheries Agreement[23]. Risk assessments of ocean acidification have been conducted for Alaska's fishery sector, and it has been suggested that southern rural areas are most at risk due to their dependence on susceptible species for nutrition and income[64]. However, the relationship between ocean acidification and its impact on fish in the central Arctic Ocean is poorly understood. Several laboratory experiments are conducted on vulnerability to acidified conditions for Arctic commercial species, such as polar cod[65] and Atlantic cod[66]. Further studies are needed to bridge the differences between laboratory experiments and ecosystems. That is, tolerance to ocean acidification, synergistic impacts by multiple stressors, and spatial and temporal variability of marine environments should be considered[67]. Furthermore, food webs in the Arctic Ocean are characterized by short linkages. Thus, the negative impacts of ocean acidification on zooplankton and calcifying benthic invertebrates could rapidly reduce the food supply for upper trophic predators[25].

The anomalous low oxygen and acidification event is likely a unique phenomenon on the CP in the Arctic high-seas. It could not occur in the eastern Arctic, as oxygen-rich AW and LWH ventilate this ocean part. Nonetheless, ocean acidification may be accelerated in the eastern Arctic due to Atlantification. The gateway area of AW inflow is a net annual ocean $CO_2$ sink, mainly caused by biological $CO_2$ uptake[68]. This effect could increase under a scenario of sea ice loss accompanied by Atlantification. In addition to the biological $CO_2$ uptake, strong winter ventilation and AW's high alkalinity contribute to the sink of

atmospheric $CO_2$ in the AW gateway area[69]. Moreover, dense $CO_2$-rich brine rejection in winter on the shallow shelf of the Barents Sea and transport of dense $CO_2$-rich water to intermediate and deep layers of the eastern Arctic could effectively increase carbon storage there[70]. As a result of this increasing carbon storage, the aragonite saturation state $\Omega_{arg}$ has decreased by 0.05–0.18 over the last two decades (1996–2015), and it will become less than 1 in the intermediate and deep layers (>~500 m) of the eastern Arctic by 2100[71]. Atlantification is likely advantageous for Atlantic cod fisheries in the eastern Arctic, but further warming and ocean acidification could lead to a steep decline in its stock and the fisheries are predicted to be at risk of collapse by 2100[72]. On the contrary, the CP was already occupied by water corrosive to aragonite ($\Omega_{arg}$ <1) in 2020, albeit intermittently. In addition, there will be a threat of deoxygenation in the future. Therefore, this site should be monitored as a bellwether of ecosystem degradation caused by ocean acidification and deoxygenation in the Arctic high-seas.

## Methods

### Ship-based sampling campaign in 2020

The Synoptic Arctic Survey, a coordinated multiship and multination pan-Arctic ship-based sampling campaign, was planned to be implemented in 2020 to obtain a synoptic view of the totality of hydrographic and ecosystem changes occurring in the Arctic Ocean[20]. However, several of these research cruises were canceled due to COVID-19. Even in such a situation, Canada/US, Japan, and Korea managed to conduct field expeditions that covered a wide area of the western Arctic Ocean (Fig. 1b). The Joint Ocean Ice Study/BG Exploration Project (Canada/US) was performed in the CB by CCGS Louis S. St-Laurent from September 14 to October 2, 2020 (see cruise report; [https://www2.whoi.edu/site/beaufortgyre/wp-content/uploads/sites/108/2021/05/2020-79-LSSL-Cruise-Report-v2021-02-23.pdf](https://www2.whoi.edu/site/beaufortgyre/wp-content/uploads/sites/108/2021/05/2020-79-LSSL-Cruise-Report-v2021-02-23.pdf)). The R/V Araon (Korea) cruise operated in the Bering Strait, Chukchi, and Eastern Siberian seas, and the CB and MB during August 2020 as a part of the project titled "Korea-Arctic Ocean Warming and Response of Ecosystem (K-AWARE)." In between the study areas by the CCGS Louis S. St-Laurent and R/V Araon, a research cruise of R/V Mirai (Japan) was carried out in October 2020 as a part of the Arctic Challenge for Sustainability II (ArCS II) project (see cruise report; [https://www.jamstec.go.jp/iace/e/report/pdf/2020.MR20-05.pdf](https://www.jamstec.go.jp/iace/e/report/pdf/2020.MR20-05.pdf)).

Hydrographic data were obtained in each cruise using an SBE 911 plus conductivity-temperature-depth (CTD) system (Sea-Bird Scientific, Bellevue, WA, USA). The CTD system was set up with temperature, conductivity (salinity), and dissolved oxygen sensors. On the R/V Mirai, a nitrate sensor (Deep SUNA; Sea-Bird Scientific, Bellevue, WA, USA) was also attached to the system. The temperature sensor's accuracy is ±0.001 °C. Sensor data for salinity, dissolved oxygen, and nitrate were corrected to be highly correlated with analytical values of water collected from Niskin bottles at the same locations and depths as the sensors.

Salinity samples from Niskin bottles were analyzed following the Global Ocean Ship-based Hydrographic Investigations Program (GO-SHIP) Repeat Hydrography Manual using a salinometer (AUTOSAL 8400B; Guildline Instruments, Smiths Falls, ON, Canada) and the International Association for the Physical Sciences of the Oceans (IAPSO) standard seawater as reference material[73]. The precision of salinity measurements was ~0.001.

Dissolved oxygen samples from Niskin bottles were measured through Winkler titration based on the World Ocean Circulation Experiment Hydrographic Program method[74]. The precision of dissolved oxygen measurements was ~0.2 $\mu$mol kg$^{-1}$.

In this study, we also used data for nutrients, total alkalinity, and dissolved inorganic carbon (DIC) obtained from the R/V Mirai to calculate the following parameters: NO, PO, NO/PO, and $\Omega_{arg}$. These parameters were calculated using Ocean Data View ([https://odv.awi.de](https://odv.awi.de))

developed by R. Schlitzer (Alfred Wegener Institute for Polar and Marine Research). Hereafter, we describe the methods adopted by the R/V Mirai cruise.

Nutrient samples were analyzed according to the GO-SHIP Repeat Hydrography Manual[75] using certified reference materials produced by KANSO TECHNOS Co., Ltd. (http://www.kanso.co.jp/eng/production/index.html). Precisions were expressed as the coefficient of variation (CV), i.e., 0.13% for nitrate, 0.18% for nitrite, 0.14% for phosphate, 0.11% for silicate, and 0.29% for ammonium.

The total alkalinity of the samples was measured following a protocol developed using a spectrophotometric system[76]. Its values were calibrated against certified reference materials from Dr. Dickson of the Scripps Institute of Oceanography. The precision of the total alkalinity measurement was 1.1 μmol kg$^{-1}$.

Seawater DIC was determined using a coulometer (Model 3000; Nihon ANS Inc., Tokyo, Japan). Data were corrected using the same certified reference materials for the total alkalinity measurements. The precision of DIC was 0.63 μmol kg$^{-1}$.

### Ship-based observations before 2020
The R/V Mirai data utilized for this study were acquired in 2002, 2004, 2008, 2009, 2010, and 2012–2020. We also used the data obtained from CCGS Louis S. St-Laurent under a collaborative framework of the BG Exploration Project that started in 2003 and has been performed yearly since then. These two datasets allow us to study interannual changes in the BG and its impact on the marine environment of the western Arctic. This study also included data from the Chukchi Borderland Project[77] and International Siberian Shelf Study[78] conducted by the United States Coast Guard Cutter Polar Star (USA) in 2002 and Yacob Smirniskyi (Russia) in 2008. Together with the R/V Araon data, this information sheds light on the dynamics of water spreading in the intersectional water region north of the ESS and the Chukchi Sea. The above data from 2002 to 2020 were compared with the 1994 Arctic Ocean Section implemented by the CCGS Louis S. St-Laurent[52]. All data utilized in this study, observation periods, and data sites are summarized in Supplementary Table 1.

As the data site of R/V Mirai in Supplementary Table 1 is temporarily closed because of a security incident, we provide the data as Supplementary Data 1.

### Historical datasets from the 1950s to the 1980s
Historical datasets of temperature and salinity were compiled in the Environmental Working Group Joint US–Russian Atlas of the Arctic Ocean[27]. Temperature and salinity data were collected from Russian and western drifting stations, ice breakers, and airborne expeditions between 1948 and 1993. The products of the datasets are gridded mean fields for decadal periods (the 1950s, 1960s, 1970s, and 1980s) and all periods from 1948 to 1993. This study used the product of the mean field from 1948 to 1993 (Nonetheless, the data were mainly obtained during the 1950s to 1980s). The Atlas is acquired from the site linked below.

https://nsidc.org/data/g01961/versions/1

Historical datasets of dissolved oxygen and other chemical parameters (e.g., nutrients) were compiled in the Hydrochemical Atlas of the Arctic Ocean[26]. Hydrochemical data were collected from multiple cruises between 1948 and 2000. Herein, we used data from 1948 to 1993, i.e., the same period as the temperature and salinity data acquisition described above (Note that the hydrochemical data we used were concentrated in a period between 1960 to 1990). The temperature and salinity of the hydrochemical data (non-gridded data) were interpolated from the gridded mean field of the Environmental Working Group Joint US–Russian Atlas of the Arctic Ocean[27]. The Hydrochemical Atlas is acquired from the site linked below.

https://doi.org/10.1594/PANGAEA.691332

### Data gridding
Spatial maps of dynamic height and oxygen saturation (on a 0.5° × 2.5° latitude–longitude grid) and vertical sections of temperature and salinity (along a band of 75–76°N with a 1.0° longitude grid) in Fig. 5 were constructed using an Optimal Interpolation (OI) method[79] for the following three periods: the 1950s–1980s, 2008–2016, and 2017–2020. For gridding of the 1950s–1980s, we processed the data from the Environmental Working Group Joint US–Russian Atlas of the Arctic Ocean[27] and the Hydrochemical Atlas of the Arctic Ocean[26]. The data from the ship-based observations listed in Supplementary Table 1 were used to create the gridded datasets for 2008–2016 and 2017–2020. Values were calculated at gridded points from averages weighted by the distance from each observation point. The weights were assumed to be proportional to the covariability of data. Based on the available data, they could be approximately represented by a Gaussian function with a length scale of 150 km. The interpolation uncertainty in each gridded point was derived from the standard deviation, covariability, and weights according to least squares procedures[79]. The estimated uncertainties for the parameters shown in Fig. 5 are displayed in Supplementary Fig. 3.

### Model experiment
A tracer experiment was conducted to visualize water transport from the shelf-slope north of the ESS using the Center for Climate System Research Ocean Component Model (COCO) version 4.9[80]. The pan-Arctic regional COCO model used in this analysis has reasonably reproduced major fields of ocean circulation in the western Arctic[81,82]. The model configuration, experimental design, and corresponding physical hydrographic/circulation results are the same as in the previous experiment[82]. The horizontal resolution is approximately 5 km, enough to represent detailed ocean features such as eddies and local upwellings[83,84]. Atmospheric forcing components were constructed from the Climate Forecast System Reanalysis version 2 (CFSv2) 6-hourly dataset of the National Centers for Environmental Prediction (NCEP)[85]. A virtual passive tracer with a concentration value of 1 (assuming the lowest oxygen saturation water) was continuously given in a water column from 100 m to the seafloor within 100–200 m isobaths and 170° E–170° W throughout the experiment period for 2013–2020. Advection and diffusion of the tracer were calculated with ocean temperature and salinity. The tracer values in the entire model domain were reset to zero once on January 1, 2017, to compare tracer distributions between the following two periods: 2013–2016 and 2017–2020 (4 years each).

## Data availability
Data related to this paper are publicly available and can be downloaded from the following data sites (see also Methods and Supplementary information). R/V Araon data from https://kpdc.kopri.re.kr/search/694ee19e-1c0b-4a44-8fd6-e83c94992731; R/V Mirai data from http://www.godac.jamstec.go.jp/darwin/e or Supplementary Data 1; BGEP data from https://www2.whoi.edu/site/beaufortgyre/data/ctd-and-geochemistry/; AOS 1994 data from https://cchdo.ucsd.edu/cruise/18SN19940724; CBL 2002 data from http://psc.apl.washington.edu/HLD/CBL/CBL.html; ISSS 2008 data from https://cchdo.ucsd.edu/cruise/90JS20080815; Environmental Working Group Joint US–Russian Atlas of the Arctic Ocean[27] from https://nsidc.org/data/g01961/versions/1; Hydrochemical Atlas of the Arctic Ocean[26] from https://doi.org/10.1594/PANGAEA.691332. Source data are provided with this paper.

## Code availability
The code of the numerical experiment is the same as that used in the previous study[82], which is based on the Center for Climate System Research Ocean Component Model (COCO)[80] found at https://ccsr.aori.u-tokyo.ac.jp/~hasumi/COCO/. The original code will be available

upon request, and necessary information can be provided through the above website or the corresponding author.

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

## Acknowledgements

We thank the officers, crews, and research technicians of the R/V Araon, R/V Mirai, and CCGS Louis S. St-Laurent. We appreciate R. Krishfield (Woods Hole Oceanographic Institution), I. A. Le Bras (Woods Hole Oceanographic Institution), A. Proshutinsky (Woods Hole Oceanographic Institution), and M.-L. Timmermans (Yale University) led the BG Exploration Project. The model experiment was conducted using the JAMSTEC Earth Simulator. Maps and figures were drawn using Ocean Data View (https://odv.awi.de) developed by R. Schlitzer (Alfred Wegener Institute for Polar and Marine Research) and Grid Analysis and Display System (GrADS; http://cola.gmu.edu/grads/). This research was partly supported by Korea Institute of Marine Science & Technology Promotion (KIMST) funded by the Ministry of Oceans and Fisheries (20210605, Korea-Arctic Ocean Warming and Response of Ecosystem (K-AWARE), KOPRI) (J.J., K.-H.C., E.J.Y., and S.-H.K.); the Arctic Challenge for Sustainability II (ArCS II), Program Grant Number JPMXD1420318865, funded by the Ministry of Education, Culture, Sports, Science and Technology of Japan (MEXT) (S.N., A.F., A.M., M.I., E.W., M.A., M.Y.-K., and T.K.); and the Joint Ocean Ice Study (JOIS), funded by Fisheries and Oceans Canada, in collaboration with the BG Exploration Project, funded by the National Science Foundation Office of Polar Programs (W.J.W., M.I., and M.Y.-K.).

## Author contributions

S.N. and J.J. conceived the study and wrote the manuscript. S.N., J.J., K.-H.C., W.J.W., A.F., A.M., M.I., M.A., M.Y.-K., and E.J.Y. collected and validated data. E.W. performed numerical experiments. T.K., E.J.Y., and S.-H.K. coordinated ship-based observation projects. S.N., J.J., K.-H. C., W.J.W., A.F., A.M., M.I., E.W., M.A., M.Y.-K., T.K., E.J.Y., and S.-H.K. contributed to reviewing and improving the manuscript.

## Competing interests

The authors declare no competing interests.
