## [Peer Review File · Nature Communications]

REVIEWER COMMENTS

Reviewer #1 (Remarks to the Author):

In this manuscript a phenomenon of reduced oxygen concentration at intermediate depth in the western Arctic Ocean observed in 2020 is described. The authors propose that the cause of this phenomenon is the reduction of the Beaufort Gyre extent relative to the status of the past two decades and the progress of the Arctic Atlantification.

The authors did a good job in analyzing their observations, but there is a major issue in the main part of the manuscript. The lower oxygen concentration at the intermediate depth in the large Canada Basin in 2020 cannot be well explained by the processes described by the authors. As shown in Fig3h, there was a big tongue of high oxygen entering the gyre from the northeast of the gyre averaged over 2008-2016. This source was absent in 2020 in Fig3g. The shift of the low oxygen area from the ESS slope to the southern part of the Chukchi Plateau cannot explain the reduction of oxygen concentration that occurred almost everywhere in the Canada Basin, although some of low oxygen water could have entered the gyre circulation from the southwest. The large part of the change in the oxygen concentration in the Canada Basin should be linked to the changes in the large-scale circulation pathways of different water masses in the Arctic Ocean, especially those that can enter the gyre from the northwestern and northern boundaries of the Beaufort Gyre. That is, the phenomenon of the shadow zone cannot explain the phenomenon of decadal changes in the Canada Basin.

Furthermore, the fact that the Beaufort Gyre extent has reduced at the end of the last decade was reported by some of the coauthors of this manuscript (Jung2021) and in other publications as well. The related novelty is not big without further dynamic analysis.

I would suggest rejection based on the above two major comments.

A few editorial comments:

L33 -> sea ice cover decreased

L40 -> increased eddy activity or numbers

L56 -> the core of

105 -> should have occurred

106 salinity of about

107 where salinity is ...

108 could be formed by brine rejection

109 or through a long-time contact of the water with the ...

119 low in NO

143 as used in previous studies and described below

151 Wind strength was also very different in the past from that in the recent decades.

178 cause the decline of sea ice in ...

180 induce stronger mixing

183-184 the impacts of sea ice decline and wind outside the Canada Basin cannot be ignored

185-196 the spreading of these water masses, especially ...

214/215 the area with lowest DO

231-234 actually there is reduction in the supply from the eastern part of the Beaufort Gyre as shown in Fig 3g,h.

How to explain the difference in the DO inside the Beaufort Gyre between the periods before and after 2000?

254 had an order of magnitude ...

256-257 it takes approximately two years for all the volume of ExLowDO water on the shelf-slope to be removed by the frontal northward flow along the Chukchi Plateau.

265 -> 20 years ago

At the end of the paper, some discussions about the significance of the impact of reduced oxygen concentration at the intermediate depth relative to the stable high oxygen concentration in the upper 200 m would be useful.

The potential impact of the “very high” concentration of oxygen in the Makarov Basin in 2020 (shown in Fig3g) was not discussed. This could be an interesting aspect.

It would be good to reduce the number of acronyms if possible (like use salinity instead of S in the text).

Reviewer #2 (Remarks to the Author):

In this manuscript entitled "Shrink of an ocean gyre in the Pacific Arctic and Atlantification open a door of shadow zone", the authors explored recent hydrographic and biogeochemical changes in the Pacific Arctic. First, I would like to emphasize that the manuscript is well structured and interesting to read, and the potential implications for the Arctic are major. The description of the mechanisms is really well done (and I don't have comments on the core and demonstration of the paper), but the discussion of the implications needs to be more elaborate. Since we are just talking about fisheries (l.84-85, l. 276-277), in such journals, it is crucial to reveal the main implications of this work, and this part, in my opinion, needs to be elaborated: for example, very briefly listed, what about acidification, what about its impact on the Arctic carbon cycle, what about the impact on other trophic levels, do you expect a similar scenario in the Barents Sea, do you expect to see a build-up of these low-oxygen waters in the central Arctic?

Specific comments:

l.1: I think the title needs to be changed. "Open a door to the shadow zone". I would not keep the term shadow zone in the title, as well as "open the door".

l.27 "shrank to the east of an ocean ridge" Do you have any idea how far or in what area this might apply?

l. 31: "towards the fishable area". Since the Canadian basin is considered a true ocean desert (in terms of productivity). Is there significant fishing effort here, do you think that is really the main implication of your results?

l.61-64. During the survey, anomalous water characteristics were observed in the boundary region. Here we examine the formation mechanism of this water by combining the data obtained by the R/V Araon (Korea), R/V Mirai (Japan), and CCGS Louis S. St-Laurent (Canada) in the SAS project. The end of this introduction and the transition could be improved, so that the paragraph does not end with a list of data, but rather with the main message of the paper.

l.83-84: "This anomalous DO and Ω water may impact marine ecosystems around the Chukchi Plateau, which is expected to be a fishable area in the future ice-free Arctic Ocean^{18,19}. Polar species are most sensitive to changes in DO₂₀". This transition needs to be improved.

Reviewer #3 (Remarks to the Author):

A. Key results:

The authors reported the apparent transport of waters with low oxygen and aragonite saturation state from the shelf into the interior Arctic ocean, and attribute this to shifts in gyre circulation. While changes in the physical oceanography of the Arctic Ocean have been previously noted, this observation is novel in demonstrating large associated changes of key biogeochemical stressors of ocean change penetrating into ocean basin. I agree with the authors that this change is potentially relevant for ecosystems and future fisheries in the Western Arctic Ocean, however such ecological implications are not explored in this manuscript.

B. Validity:

The data interpretation about the source of the water mass and reason for frontal formation and enhanced transport is plausible. This analysis relies on a comparison of hydrographic conditions during one well sampled year with two coarsely binned prior periods. This comparative approach necessarily limits how robustly the authors can infer which underlying mechanisms drive the observations. One possible extension of the work in the discussion would be to suggest sensitivity studies or numerical model experiments that might be used to test their hypothesis.

Alternative explanations that are mentioned could be considered in more detail in this manuscript as well (e.g., frontal formation was previously limited because of the smaller role of Atlantic water mass penetration into the Eastern Arctic Ocean, an increased respiratory signature might also derive from changing export or erosion)--I expect that the evidence at hand from the hydrographic datasets assembled and literature cited by the authors is enough to make some statement as to which of these are plausible or likely, or to say what additional information should be gathered moving forward.

I have three main concerns about the validity of the analysis:

(1) Figures 2 and 3 indicate that the the spatial sampling of the low-oxygen regions is sparse. No spatial or temporal statistics are presented about the robustness and significance of the change in oxygen shadow zone between periods. Because sampling effort varied in time and space, I think it is also reasonable to examine to what degree features may become homogenized or variance overestimated because of the sampling approach. Specifically, in Figure 3 it looks like the length of sampling period considered may be correlated with the spatial smoothing in dynamic height and the sections, while the spatial clustering of oxygen sampling in the oldest time period could be a driver of the stronger spatial variance in DO. If there are not compelling spatial and temporal analyses that could address these questions, the authors could provide a caveat about factors related to variable sampling effort across the compared times and locations.

(2) Similarly, it is unclear if the difference in biogeochemistry is associated with a shift in the state in the system or a seasonal/interannual/interdecadal pattern--the older comparison periods appear to include a slightly wider annual range of dates than the 2020 sampling (Table S1). The authors could perhaps exclude this possibility by examining climatological seasons or climate indices across all data in the supplement. I don't think the answer to spatial or temporal analyses of these sorts will undercut the importance of the result (that gyre circulation can enhance the offshore transport of ocean change stressors), but it could lead to different conclusions about trends.

(3) Finally, the estimates regarding timescales of respiration and transport in the Discussion seem predicated on the assumption that low-oxygen conditions are relocated from one location to another. I expected that greater transport would spread and weaken the signal, rather than move it wholesale (the shelf sediments remain the primary location of respiration, correct?). These different assumptions might be reconciled with simple calculations or numerical experiments involving an advection-diffusion-sink equation, instead of advection alone. The current presentation is a useful thought experiment, and I think it would be more clear if presented as such ("in a hypothetical case where the shelf sediment respiration signal is completely erased by advective transport,...").

I acknowledge that this is a complex system and the constraints of observational oceanography limit what analyses are reasonable.

C. Significance:

I think this is a potentially significant observation for Arctic oceanography. It would be more significant if (1) paired with numerical modeling or sensitivity studies to support the proposed mechanism (enhanced frontal transport of biogeochemical anomalies from the shelf) and (2) the Discussion briefly discussed potential ecological/fisheries consequences in a more quantitative or systematic manner (more than the current, single sentence). For example, the introduction noted that polar fish species may be particularly impacted by decreased oxygen; what sort of reduction in habitat or increase in physiological stress would this reduction in oxygen correspond to for a typical polar species? Or, how might the presence of lower oxygen lead to different/inequitable outcomes for signatories of the noted Arctic fisheries agreement (which needs a reference, by the way)?

D. Data and methodology:

I am satisfied that this is high quality hydrographic data that has been appropriately QA/QCed. There are a couple problems with data access: (1) I cannot easily locate the RV Araon data at the specified website--a more specific website or access instructions should be provided. (2) The historical Arctic temperature and salinity dataset is not immediately accessible at the provided link (it redirects to a

missing page). If a working permanent url is not available, please include these data in the supplement.

E. Analytical approach

The analytical approach could be strengthened by including statistical tests to back up statements like "very weak" or "very sluggish" currents, or "significant" decreases in DO or changes in Atlantic influence, and to examine the spatiotemporal aspects of the comparisons of different datasets.

In general, I think uncertainty bounds should be presented along with parameters like concentration and concentration change, current velocity, saturation state.

Uncertainty bounds are included in the ± 1 standard deviation errorbars in Figure 2b, but it is not clear from the presentation whether that is appropriate or sufficient: (1) The number of samples per year seems variable; please note the number of samples ($n=\#$) with each year. (2) Given the higher sampling effort, is not clear from this panel alone that 2020 represents a significant difference from prior years. A statistical comparison between years could help increase reader confidence that the differences are meaningful (e.g., ANOVA, k-means clustering, etc.--on appropriately transformed data to meet the test assumptions about the underlying distributions). (3) Based on the spatial variability in the specified region, it is not obvious that the DO should be normally distributed. If not, 95% confidence intervals (likely asymmetric) or box and whisker plots could provide more accurate error estimates that help readers better understand which differences are significant.

F. Suggested improvements:

Please see section C ('Significance') for ideas to improve the significance of the manuscript by extending beyond reporting a novel observation. (1) The authors' hypotheses about the transport of low-oxygen waters would be more compelling if paired with numerical model output, or a simple sensitivity study to support the favored explanation and rule out competing explanations. e.g., download some Arctic ocean model output that can resolve more periods or trends in Beaufort Gyre location and size and evaluate whether the physical transport changes are consistent with the authors' hypotheses. That may turn out to be beyond the scope of a revision, in which case the Discussion could lay these out as next steps to build on this manuscript. (2) I think that the potential ecological and fisheries consequences of this observation are very interesting (I assume the authors do too since they mentioned this three times!), and are worth more discussion. This could even include first-order quantitative estimates of potential impacts and remain well within the scope of a revision.

G. Clarity and context:

I suggest the authors use density directly (density appears to be measured or calculable for the various hydrographic datasets) instead of salinity as a proxy for density (e.g., Line 207) for the analysis and in many figures. If there is a compelling reason to use salinity as a coordinate system instead, please present it.

To simplify the presentation and figure, I suggest presenting only aragonite saturation state. Calcite can be presented in the supplement if there is a biologically or chemically compelling reason to consider it as well, but right now the manuscript is just using both as evidence of a respiration driven anomaly.

The selection of 200 umolal as an oxygen concentration threshold feels arbitrary (to me, but I appreciate the appeal of molal units). I think that it will not be obvious to many readers why this qualifies as "extremely low," given that this concentration is not atypical in other ocean basins. Instead, I suggest choosing a saturation state threshold (and plotting the figures using O₂% saturation after appropriately correcting for in situ pressure). The resulting values (~50% saturation) are more clearly low (though not "extremely low"), and the potentially dramatic implications for animals and ecosystems more obvious. This also is more comparable to the carbonate system saturation states.

There are many acronyms for readers to remember (or look up again), which slows the narrative flow. I encourage the authors to avoid as many acronyms as possible. For example, low-oxygen is hardly longer than 'ExLowDO' and easier to read.

I think that the section 'Origin of the anomalous water' could be significantly shortened. To me, this can be condensed to "The biogeochemical characteristics of the anomalously low-oxygen water are similar to those previously observed on the East Siberian shelf. [...Comparison details of O₂, Omega, NO, NO/PO...]. Thus we hypothesize that the likely source of this water mass is transport from the East Siberian shelf. Next, we detail how observed changes in Arctic circulation are consistent with this hypothesis." That could be done in several sentences instead of a page, and some of the more detailed context or background could be shifted to the supplement. I encourage the authors to consider how they might similarly streamline the following results sections to make for a short and compelling read.

In Figure 4e, it isn't clear to me why the observations are so sparse (and interpolation so patchy) given the density of sampling shown in panels c and d. Perhaps larger spatial/property bins should be chosen when matching locations/parameters to difference across time. Or in this case it seems defensible to plot the difference in the already gridded and interpolated fields from the previous

panels (with associated gridded salinity--or density--fields) and note that in the caption. That may be slightly further from the raw data, but I think it would be more illustrative.

Is there a way to introduce Figure 5 earlier in the manuscript (e.g., as Figure 2)? The text was clearer to me when rereading it after having seen Figure 5.

F. References:

Given the journal limits on citation, this is a reasonable suite of foundational and recent references.

G. Your expertise:

I am not an expert in the physical oceanography of the Arctic Ocean. Thus I am not completely confident as to whether the changes in circulation proposed are reasonable drivers of the biogeochemical changes observed. I concur with the authors that the hydrographic sections are suggestive, but would feel more confident in this interpretation if the changes in water mass volume and flow were clearly labeled/linked to the analysis as part of another section of supplemental text (perhaps difference/anomaly plots for salinity, temperature, density, and water transport as well?).

H. Final thoughts

I enjoyed this manuscript and learned from it. I think it is appropriate in scope and novelty for Nature Communications. I suggest minor revisions for clarity and increased significance.

Responses to comments of Reviewer #1 on “Shrink of an ocean gyre in the Pacific Arctic and Atlantification open a door of shadow zone” by S. Nishino, J. Jung, K.-H. Cho, W. J. Williams, A. Fujiwara, A. Murata, M. Itoh, M. Aoyama, M. Yamamoto-Kawai, T. Kikuchi, E. J. Yang, and S.-H. Kang (Paper # NCOMMS-22-30054-T).

We deeply appreciate valuable comments of the reviewer, which improved our paper significantly. We have revised the manuscript in line with suggestions from the reviewer. Our respective responses are shown below.

Reviewer #1 (Remarks to the Author):

In this manuscript a phenomenon of reduced oxygen concentration at intermediate depth in the western Arctic Ocean observed in 2020 is described. The authors propose that the cause of this phenomenon is the reduction of the Beaufort Gyre extent relative to the status of the past two decades and the progress of the Arctic Atlantification.

The authors did a good job in analyzing their observations, but there is a major issue in the main part of the manuscript. The lower oxygen concentration at the intermediate depth in the large Canada Basin in 2020 cannot be well explained by the processes described by the authors. As shown in Fig3h, there was a big tongue of high oxygen entering the gyre from the northeast of the gyre averaged over 2008-2016. This source was absent in 2020 in Fig3g. The shift of the low oxygen area from the ESS slope to the southern part of the Chukchi Plateau cannot explain the reduction of oxygen concentration that occurred almost everywhere in the Canada Basin, although some of low oxygen water could have entered the gyre circulation from the southwest. The large part of the change in the oxygen concentration in the Canada Basin should be linked to the changes in the large-scale circulation pathways of different water masses in the Arctic Ocean, especially those that can enter the gyre from the northwestern and northern boundaries of the Beaufort Gyre. That is, the phenomenon of the shadow zone cannot explain the phenomenon of decadal changes in the Canada Basin.

Response:

We agree with the reviewer. We estimated the horizontal and vertical diffusion of oxygen on the Chukchi Plateau and found that the low oxygen water observed there in 2020 would be rapidly diluted by the surrounding water through horizontal diffusion. Therefore, the low oxygen water via the Chukchi Plateau could not contribute to the decrease of oxygen in the entire Canada Basin. The low oxygen in the Canada Basin in 2020 would be attributed to an absence of high oxygen water from the northwestern and northern boundaries of the Beaufort Gyre as the reviewer pointed out. These points were described in lines 275-287 in the revised manuscript.

Furthermore, the fact that the Beaufort Gyre extent has reduced at the end of the last decade was reported by some of the coauthors of this manuscript (Jung2021) and in other publications as well. The related novelty is not big without further dynamic analysis.

Response:

Jung et al. (2021) studied a shoaling of nutricline west of the Chukchi Plateau and its impact on the phytoplankton distribution. Our study focused on the unusually low oxygen and acidified water on the Chukchi Plateau and its formation mechanism. The appearance of this unusual water has never studied before. In the revised manuscript, we clarified this point in the Introduction section. The proposed mechanism was also verified by our numerical model. For details, please see the subsection “Modeled transport of the lowest oxygen saturation water.”

A few editorial comments:

Response:

We revised the manuscript according to the reviewer’s editorial comments. Please note that some sentences containing words that were pointed out by the reviewer have been deleted or changed in the revision process.

L33 -> sea ice cover decreased

Response: Corrected.

L40 -> increased eddy activity or numbers

Response: Corrected.

L56 -> the core of

Response: Corrected.

105 -> should have occurred

Response: Corrected.

106 salinity of about

Response: Corrected, but moved to Supplementary Discussion 1.

107 where salinity is ...

Response: Corrected, but moved to Supplementary Discussion 1.

108 could be formed by brine rejection

Response: Corrected, but moved to Supplementary Discussion 1.

109 or through a long-time contact of the water with the ...

Response: Corrected, but moved to Supplementary Discussion 1.

119 low in NO

Response: Corrected, but moved to Supplementary Discussion 1.

143 as used in previous studies and described below

Response: Corrected.

151 Wind strength was also very different in the past from that in the recent decades.

Response: The following sentence was added to lines 192-193.

“despite an era of strong Beaufort High (1948–88) compared with the recent decades (ref. 25).”

178 cause the decline of sea ice in ...

Response: Corrected.

180 induce stronger mixing

Response: Corrected.

183-184 the impacts of sea ice decline and wind outside the Canada Basin cannot be ignored

Response: The following sentence was added to lines 221-223.

“The eastward penetrations of LHW and AW were presumably associated with the intensification of AW supply induced by the sea-ice decline (ref. 10) and cyclonic atmospheric circulation (ref. 25) as well as the collapse of the Beaufort High (ref. 24).”

185-196 the spreading of these water masses, especially ...

Response: Corrected.

214/215 the area with lowest DO

Response: Corrected.

231-234 actually there is reduction in the supply from the eastern part of the Beaufort Gyre as shown in Fig 3g,h. How to explain the difference in the DO inside the Beaufort Gyre between the periods before and after 2000?

Response: We agree with the reviewer as mentioned above. This point was described in lines 275-287 in the revised manuscript.

254 had an order of magnitude ...

Response: This sentence was deleted.

256-257 it takes approximately two years for all the volume of ExLowDO water on the shelf-slope to be removed by the frontal northward flow along the Chukchi Plateau.

Response: This sentence was changed as follows.

“it takes approximately one year for all the volume of the lowest oxygen saturation water on the shelf-slope to be removed by the frontal northward flow along the Chukchi Plateau.”

265 -> 20 years ago

Response: We used the term “20 years earlier.” The reason for this is that we compared the results of observations conducted in 2014 with those made 20 years earlier.

At the end of the paper, some discussions about the significance of the impact of reduced oxygen concentration at the intermediate depth relative to the stable high oxygen concentration in the upper 200 m would be useful.

Response: We estimated the horizontal and vertical diffusion of oxygen on the Chukchi Plateau and found that the low oxygen water observed there in 2020 would be rapidly diluted by the surrounding water through horizontal diffusion, which was an order of magnitude larger than vertical diffusion. Thus, the impact of low oxygen water on the upper layer seems to be small. This point was described in lines 268-274.

The potential impact of the “very high” concentration of oxygen in the Makarov Basin in 2020 (shown in Fig3g) was not discussed. This could be an interesting aspect.

Response: The high oxygen in the Makarov Basin would be a result of the penetration of LHW with high oxygen to the Makarov Basin via the Lomonosov and Mendeleev Ridges. This point was described in lines 285-287. We do not have any idea about the impact of high oxygen water on the upper layer in the Makarov Basin because its vertical oxygen gradient is much weaker than that of the low oxygen water on the Chukchi Plateau.

It would be good to reduce the number of acronyms if possible (like use salinity instead of S in the text).

Response: We used acronyms as little as possible.

We thank the editor and reviewers very much for their kindness and generosity with their time.

Responses to comments of Reviewer #2 on “Shrink of an ocean gyre in the Pacific Arctic and Atlantification open a door of shadow zone” by S. Nishino, J. Jung, K.-H. Cho, W. J. Williams, A. Fujiwara, A. Murata, M. Itoh, M. Aoyama, M. Yamamoto-Kawai, T. Kikuchi, E. J. Yang, and S.-H. Kang (Paper # NCOMMS-22-30054-T).

We deeply appreciate valuable comments of the reviewer, which improved our paper significantly. We have revised the manuscript in line with suggestions from the reviewer. Our respective responses are shown below.

Reviewer #2 (Remarks to the Author):

In this manuscript entitled "Shrink of an ocean gyre in the Pacific Arctic and Atlantification open a door of shadow zone", the authors explored recent hydrographic and biogeochemical changes in the Pacific Arctic. First, I would like to emphasize that the manuscript is well structured and interesting to read, and the potential implications for the Arctic are major. The description of the mechanisms is really well done (and I don't have comments on the core and demonstration of the paper), but the discussion of the implications needs to be more elaborate. Since we are just talking about fisheries (1.84-85, 1.276-277), in such journals, it is crucial to reveal the main implications of this work, and this part, in my opinion, needs to be elaborated: for example, very briefly listed, what about acidification, what about its impact on the Arctic carbon cycle, what about the impact on other trophic levels, do you expect a similar scenario in the Barents Sea, do you expect to see a build-up of these low-oxygen waters in the central Arctic?

Response:

Implications of low oxygen and acidified water for the Arctic marine ecosystem and fisheries agreement were additionally described in the Discussion section. Please see the Discussion section for details. Here, we extract some sentences from the Discussion section in response to the reviewer's comments above.

- Regarding ocean acidification, it is one of prioritized indicators associated with the central Arctic Ocean fisheries agreement. The Chukchi Plateau was already occupied by water corrosive to aragonite ($\Omega_{\text{arg}} < 1$) in 2020. In addition, there will be a threat of deoxygenation in the future. Therefore, this site should be monitored as a bellwether of ecosystem degradation caused by ocean acidification and deoxygenation in the Arctic Ocean. (lines 415-418)
- As for the Arctic carbon cycle, the carbon mineralization (oxygen uptake) rate might increase due to increased organic matter inputs from rivers, coastal erosion, and biological production, resulting in the formation of the lowest oxygen saturation water more quickly and widely. (lines 359-361)
- Food webs in the Arctic Ocean are characterized by short linkages; and thus, the negative

impacts of ocean acidification on zooplankton and calcifying benthic invertebrates could rapidly lead to a reduction in food supply for upper trophic predators. (lines 396-399)
 - The anomalous event of low oxygen and acidification studied in this paper is likely a unique phenomenon on the Chukchi Plateau and would not occur in the eastern Arctic (including the Barents Sea). This is because the eastern Arctic is ventilated by the oxygen-rich Atlantic Water (AW) and Lower Halocline Water (LHW). Nonetheless, ocean acidification may be accelerated in the eastern Arctic as a result of Atlantification. (lines 400-403)

Specific comments:

l.1: I think the title needs to be changed. "Open a door to the shadow zone". I would not keep the term shadow zone in the title, as well as "open the door".

Response:

We changed the title as follows.

"Beaufort Gyre shrinkage and Atlantification induced an anomalous biogeochemical event in the western Arctic Ocean"

l.27 "shrank to the east of an ocean ridge" Do you have any idea how far or in what area this might apply?

Response:

The following sentence was added to lines 187-188.

"In other words, a shrink of the Beaufort Gyre from 2008–2016 to 2017–2020 occurred, ranging from the south to north of the Chukchi Plateau."

l. 31: "towards the fishable area". Since the Canadian basin is considered a true ocean desert (in terms of productivity). Is there significant fishing effort here, do you think that is really the main implication of your results?

Response:

The following sentences were added to lines 372-379.

"The Chukchi Borderland is a potential spawning area for Arctic cod, and polar cod may distribute south of the Chukchi Borderland (ref. 18). Both are important cod species in Arctic marine food webs and are of commercial interest. Therefore, when introducing appropriate ecosystem-based management under the central Arctic Ocean fisheries agreement (www.fao.org/faolex/results/details/en/c/LEX-FAOC199323) that has recently entered into force, it will become important to monitor the marine environment and ecosystem in this region, especially across the Chukchi Plateau where anomalous events of low oxygen water and acidification are expected to increase in the future."

l.61-64. During the survey, anomalous water characteristics were observed in the boundary region. Here we examine the formation mechanism of this water by combining the data obtained by the R/V Araon (Korea), R/V Mirai (Japan), and CCGS Louis S. St-Laurent (Canada) in the SAS project. The end of this introduction and the transition could be improved, so that the paragraph does not end with a list of data, but rather with the main message of the paper.

Response:

The following sentences were added to lines 66-69.

“The proposed mechanism (Fig. 1c–d) brings new perspectives to ecosystem assessments in the Arctic Ocean (ref. 18); for example, mapping and monitoring this unusual water and its impact on the marine ecosystem. This information will ultimately help to develop policies facilitating effective management, e.g., for potential fisheries in the future ice-free Arctic Ocean.”

l.83-84: “This anomalous DO and Ω water may impact marine ecosystems around the Chukchi Plateau, which is expected to be a fishable area in the future ice-free Arctic Ocean^{18,19}. Polar species are most sensitive to changes in DO₂₀”. This transition needs to be improved.

Response:

The above sentences were moved to the Discussion section for more detailed description. Please see the third and fourth paragraphs of the Discussion section (lines 370-399).

We thank the editor and reviewers very much for their kindness and generosity with their time.

Responses to comments of Reviewer #3 on “Shrink of an ocean gyre in the Pacific Arctic and Atlantification open a door of shadow zone” by S. Nishino, J. Jung, K.-H. Cho, W. J. Williams, A. Fujiwara, A. Murata, M. Itoh, M. Aoyama, M. Yamamoto-Kawai, T. Kikuchi, E. J. Yang, and S.-H. Kang (Paper # NCOMMS-22-30054-T).

We deeply appreciate valuable comments of the reviewer, which improved our paper significantly. We have revised the manuscript in line with suggestions from the reviewer. Our respective responses are shown below.

Reviewer #3 (Remarks to the Author):

A. Key results:

The authors reported the apparent transport of waters with low oxygen and aragonite saturation state from the shelf into the interior Arctic ocean, and attribute this to shifts in gyre circulation. While changes in the physical oceanography of the Arctic Ocean have been previously noted, this observation is novel in demonstrating large associated changes of key biogeochemical stressors of ocean change penetrating into ocean basin. I agree with the authors that this change is potentially relevant for ecosystems and future fisheries in the Western Arctic Ocean, however such ecological implications are not explored in this manuscript.

Response:

Implications of low oxygen and acidified water for the Arctic marine ecosystem were additionally described in the Discussion section.

B. Validity:

The data interpretation about the source of the water mass and reason for frontal formation and enhanced transport is plausible. This analysis relies on a comparison of hydrographic conditions during one well sampled year with two coarsely binned prior periods. This comparative approach necessarily limits how robustly the authors can infer which underlying mechanisms drive the observations. One possible extension of the work in the discussion would be to suggest sensitivity studies or numerical model experiments that might be used to test their hypothesis.

Response:

To test our hypothesis, we conducted numerical experiments. The results were described in the subsection “Modeled transport of the lowest oxygen saturation water.” We also created grid datasets using an optimal interpolation method with interpolation errors. The validity of the data interpretation was discussed in terms of the errors (see Supplementary

Discussion 2).

Alternative explanations that are mentioned could be considered in more detail in this manuscript as well (e.g., frontal formation was previously limited because of the smaller role of Atlantic water mass penetration into the Eastern Arctic Ocean, an increased respiratory signature might also derive from changing export or erosion)--I expect that the evidence at hand from the hydrographic datasets assembled and literature cited by the authors is enough to make some statement as to which of these are plausible or likely, or to say what additional information should be gathered moving forward.

Response:

The previous limited frontal formation due to a smaller role of Atlantification was additionally explained in the context of the depths of salinity = 34 and 34.5, which are the upper boundaries of Lower Halocline Water (LHW) and Atlantic Water (AW), respectively (see lines 227-230). Likely sources of organic matters associated with the respiratory signature were additionally discussed in the context of rivers, coastal erosion, and biological production (see lines 359-365).

I have three main concerns about the validity of the analysis:

(1) Figures 2 and 3 indicate that the spatial sampling of the low-oxygen regions is sparse. No spatial or temporal statistics are presented about the robustness and significance of the change in oxygen shadow zone between periods. Because sampling effort varied in time and space, I think it is also reasonable to examine to what degree features may become homogenized or variance overestimated because of the sampling approach. Specifically, in Figure 3 it looks like the length of sampling period considered may be correlated with the spatial smoothing in dynamic height and the sections, while the spatial clustering of oxygen sampling in the oldest time period could be a driver of the stronger spatial variance in DO. If there are not compelling spatial and temporal analyses that could address these questions, the authors could provide a caveat about factors related to variable sampling effort across the compared times and locations.

Response:

As the reviewer pointed out, the observation points in each period are not spatially uniform, and comparisons between hydrographic conditions with different data spacings may lead to a distortional feature. Thus, the observational data were interpolated to grid points in each period for the dynamic height (Fig. 4a–c), temperature and salinity sections (Fig. 4d–f), and oxygen saturation distributions (Fig. 4g–i). The interpolation errors inherent to each parameter were also estimated (Supplementary Discussion 2). Please note that the periods in the revised manuscript were set to be 2017–2020, 2008–2016, and 1948–1993. Figure 3 in the previous manuscript was replaced by Fig. 4 in the revised manuscript. As for Fig. 2 (oxygen distribution), we did not show grid data because we would like to focus on the oxygen distribution observed in 2020.

(2) Similarly, it is unclear if the difference in biogeochemistry is associated with a shift in the state in the system or a seasonal/interannual/interdecadal pattern--the older comparison periods appear to include a slightly wider annual range of dates than the 2020 sampling (Table S1). The authors could perhaps exclude this possibility by examining climatological seasons or climate indices across all data in the supplement. I don't think the answer to spatial or temporal analyses of these sorts will undercut the importance of the result (that gyre circulation can enhance the offshore transport of ocean change stressors), but it could lead to different conclusions about trends.

Response:

Only the observed data listed in the Supplementary Table 1, it is difficult to describe the seasonality of the anomalous water on the Chukchi Plateau. Thus, this point was examined using our numerical model. The model shows that the appearance of the anomalous water is not continuous throughout a year; rather, it was intermittent on monthly timescales with no seasonality. Please see the subsection "Modeled transport of the lowest oxygen saturation water" for details.

(3) Finally, the estimates regarding timescales of respiration and transport in the Discussion seem predicated on the assumption that low-oxygen conditions are relocated from one location to another. I expected that greater transport would spread and weaken the signal, rather than move it wholesale (the shelf sediments remain the primary location of respiration, correct?). These different assumptions might be reconciled with simple calculations or numerical experiments involving an advection-diffusion-sink equation, instead of advection alone. The current presentation is a useful thought experiment, and I think it would be more clear if presented as such ("in a hypothetical case where the shelf sediment respiration signal is completely erased by advective transport,...").

Response:

Because the biogeochemical characteristics of low oxygen water between the ESS shelf-slope and the Chukchi Plateau were almost the same (see Supplementary Discussion 1), the water would not be largely modified by mixing with the surrounding water during its migration to the Chukchi Plateau via the shelf-slope. However, after leaving the Chukchi Plateau by the frontal northward flow, the water was largely modified. We estimated horizontal and vertical diffusion from the observed data, and found that the oxygen concentration increased primarily through horizontal diffusion, which was an order of magnitude larger than vertical diffusion. These points were described in lines 264-274.

C. Significance:

I think this is a potentially significant observation for Arctic oceanography. It would be more significant if (1) paired with numerical modeling or sensitivity studies to support the

proposed mechanism (enhanced frontal transport of biogeochemical anomalies from the shelf) and (2) the Discussion briefly discussed potential ecological/fisheries consequences in a more quantitative or systematic manner (more than the current, single sentence). For example, the introduction noted that polar fish species may be particularly impacted by decreased oxygen; what sort of reduction in habitat or increase in physiological stress would this reduction in oxygen correspond to for a typical polar species? Or, how might the presence of lower oxygen lead to different/inequitable outcomes for signatories of the noted Arctic fisheries agreement (which needs a reference, by the way)?

Response:

Regarding (1), we conducted numerical experiments to verify the proposed mechanism. Please see the subsection “Modeled transport of the lowest oxygen saturation water” for details. We also created grid datasets using an optimal interpolation method with interpolation errors. The validity of the data interpretation was discussed in terms of the errors (see Supplementary Discussion 2).

As for (2), implications of low oxygen and acidified water for the Arctic marine ecosystem and fisheries agreement (www.fao.org/faolex/results/details/en/c/LEX-FAOC199323) were additionally described in the Discussion section. Please see the section for details.

D. Data and methodology:

I am satisfied that this is high quality hydrographic data that has been appropriately QA/QCed. There are a couple problems with data access: (1) I cannot easily locate the RV Araon data at the specified website--a more specific website or access instructions should be provided. (2) The historical Arctic temperature and salinity dataset is not immediately accessible at the provided link (it redirects to a missing page). If a working permanent url is not available, please include these data in the supplement.

Response:

Regarding (1), we listed a more specific website (see, for example, Supplementary Table 1).

As for (2), we are sorry for listing an inaccessible site. Now, a working permanent url is listed.

E. Analytical approach

The analytical approach could be strengthened by including statistical tests to back up statements like "very weak" or "very sluggish" currents, or "significant" decreases in DO or changes in Atlantic influence, and to examine the spatiotemporal aspects of the comparisons of different datasets.

In general, I think uncertainty bounds should be presented along with parameters like concentration and concentration change, current velocity, saturation state.

Response:

Please forgive the repetitive explanations. The observational data were interpolated to grid points in each period for the dynamic height (Fig. 4a–c), temperature and salinity sections (Fig. 4d–f), and oxygen saturation distributions (Fig. 4g–i) using an optimal interpolation method with interpolation errors. The validity of the data interpretation was discussed in terms of the errors (see Supplementary Discussion 2).

Uncertainty bounds are included in the ± 1 standard deviation error bars in Figure 2b, but it is not clear from the presentation whether that is appropriate or sufficient: (1) The number of samples per year seems variable; please note the number of samples ($n=\#$) with each year. (2) Given the higher sampling effort, is not clear from this panel alone that 2020 represents a significant difference from prior years. A statistical comparison between years could help increase reader confidence that the differences are meaningful (e.g., ANOVA, k-means clustering, etc.--on appropriately transformed data to meet the test assumptions about the underlying distributions). (3) Based on the spatial variability in the specified region, it is not obvious that the DO should be normally distributed. If not, 95% confidence intervals (likely asymmetric) or box and whisker plots could provide more accurate error estimates that help readers better understand which differences are significant.

Response:

Regarding the temporal change in oxygen (Fig. 2c in the revised manuscript), we are interested in how anomalous the levels of low oxygen water are in 2020. Therefore, we replaced the figure with standard deviations by box and whisker plots. Please see Fig. 2c and lines 101-108 for details.

F: Suggested improvements:

Please see section C ('Significance') for ideas to improve the significance of the manuscript by extending beyond reporting a novel observation. (1) The authors' hypotheses about the transport of low-oxygen waters would be more compelling if paired with numerical model output, or a simple sensitivity study to support the favored explanation and rule out competing explanations. e.g., download some Arctic ocean model output that can resolve more periods or trends in Beaufort Gyre location and size and evaluate whether the physical transport changes are consistent with the authors' hypotheses. That may turn out to be beyond the scope of a revision, in which case the Discussion could lay these out as next steps to build on this manuscript. (2) I think that the potential ecological and fisheries consequences of this observation are very interesting (I assume the authors do too since they mentioned this three times!), and are worth more discussion. This could even include first-order quantitative estimates of potential impacts and remain well within the scope of a revision.

Response:

Regarding (1), we conducted numerical experiments. The results were described in the

subsection “Modeled transport of the lowest oxygen saturation water.”

As for (2), implications of low oxygen and acidified water for the Arctic marine ecosystem and fisheries agreement (www.fao.org/faolex/results/details/en/c/LEX-FAOC199323) were additionally described in the Discussion section. Please see the section for details.

G. Clarity and context:

I suggest the authors use density directly (density appears to be measured or calculable for the various hydrographic datasets) instead of salinity as a proxy for density (e.g., Line 207) for the analysis and in many figures. If there is a compelling reason to use salinity as a coordinate system instead, please present it.

Response:

In the revised manuscript, we used density in Fig 4g–i to examine the oxygen saturation distributions along isopycnal surfaces. However, water masses in the Arctic Ocean are generally characterized by salinity in the previous studies. Thus, we used salinity when we defined water masses and drew figures (e.g., vertical sections) associated with the water masses. The following description was added to lines 141-142.

“In the cold Arctic seas, salinity mainly determines the water density and is used to identify water masses...”

To simplify the presentation and figure, I suggest presenting only aragonite saturation state. Calcite can be presented in the supplement if there is a biologically or chemically compelling reason to consider it as well, but right now the manuscript is just using both as evidence of a respiration driven anomaly.

Response:

As the reviewer pointed out, the calcite saturation state does not advance this study. So, we deleted the description and figure regarding the calcite saturation state.

The selection of 200 umolal as an oxygen concentration threshold feels arbitrary (to me, but I appreciate the appeal of molal units). I think that it will not be obvious to many readers why this qualifies as "extremely low," given that this concentration is not atypical in other ocean basins. Instead, I suggest choosing a saturation state threshold (and plotting the figures using O₂% saturation after appropriately correcting for in situ pressure). The resulting values (~50% saturation) are more clearly low (though not "extremely low"), and the potentially dramatic implications for animals and ecosystems more obvious. This also is more comparable to the carbonate system saturation states.

Response:

According to the reviewer comments, we did not use a term “extremely low” oxygen.

Instead of the oxygen concentration, we used the oxygen saturation and set a threshold to 50 % saturation. As described in lines 104-107, water with oxygen saturation levels less than 50 % was found in 2020 and it had never been observed before at least in the last two decades (Fig. 2c).

There are many acronyms for readers to remember (or look up again), which slows the narrative flow. I encourage the authors to avoid as many acronyms as possible. For example, low-oxygen is hardly longer than 'ExLowDO' and easier to read.

Response:

We used acronyms as little as possible.

I think that the section 'Origin of the anomalous water' could be significantly shortened. To me, this can be condensed to "The biogeochemical characteristics of the anomalously low-oxygen water are similar to those previously observed on the East Siberian shelf. [...Comparison details of O₂, Omega, NO, NO/PO...]. Thus we hypothesize that the likely source of this water mass is transport from the East Siberian shelf. Next, we detail how observed changes in Arctic circulation are consistent with this hypothesis." That could be done in several sentences instead of a page, and some of the more detailed context or background could be shifted to the supplement. I encourage the authors to consider how they might similarly streamline the following results sections to make for a short and compelling read.

Response:

Comparison details of O₂, Omega, NO, NO/PO were moved to Supplementary Discussion 1. We also rearranged the following subsection "Changes in PW and AW circulation" so that our hypothesis and experimental setup were described at the beginning.

In Figure 4e, it isn't clear to me why the observations are so sparse (and interpolation so patchy) given the density of sampling shown in panels c and d. Perhaps larger spatial/property bins should be chosen when matching locations/parameters to difference across time. Or in this case it seems defensible to plot the difference in the already gridded and interpolated fields from the previous panels (with associated gridded salinity--or density--fields) and note that in the caption. That may be slightly further from the raw data, but I think it would be more illustrative.

Response:

We deleted Fig. 4 (which was shown in the previous manuscript) and the related description based on a comment from another reviewer. In the previous manuscript, the decrease of oxygen in the Canada Basin shown in Fig. 4 was attributed to the spreading of low oxygen water from the Chukchi Plateau. However, as estimated in the revised manuscript, the low

oxygen water would be rapidly diluted by the surrounding water through horizontal diffusion north of the Chukchi Plateau, and therefore, could not contribute to the decrease of oxygen in the entire Canada Basin. For details, please see the last paragraph of the subsection “Oxygen saturation distribution determined by ocean circulation.”

Is there a way to introduce Figure 5 earlier in the manuscript (e.g., as Figure 2)? The text was clearer to me when rereading it after having seen Figure 5.

Response:

Figure 5 was combined with Fig. 1. This should give the reader an initial understanding of our hypothesis.

G. Your expertise:

I am not an expert in the physical oceanography of the Arctic Ocean. Thus I am not completely confident as to whether the changes in circulation proposed are reasonable drivers of the biogeochemical changes observed. I concur with the authors that the hydrographic sections are suggestive, but would feel more confident in this interpretation if the changes in water mass volume and flow were clearly labeled/linked to the analysis as part of another section of supplemental text (perhaps difference/anomaly plots for salinity, temperature, density, and water transport as well?).

Response:

We calculated the differences of dynamic height, salinity, and oxygen saturation distributions between 2017–2020 and 2008–2016, and showed them in Supplementary Fig. 4. The difference of dynamic height (Supplementary Fig. 4a) clearly indicates the penetration of LHW to the Chukchi Plateau. The difference of salinity section (Supplementary Fig. 4b) suggests shoaling of halocline that is consistent with the LHW penetration. The LHW penetration and the accompanying northward flow along the Chukchi Plateau could increase (decrease) the oxygen saturation along the shelf slope of the East Siberian Sea (along the Chukchi Plateau) as shown in Supplementary Fig. 4c. Please see Supplementary Discussion 2 for details.

H. Final thoughts

I enjoyed this manuscript and learned from it. I think it is appropriate in scope and novelty for Nature Communications. I suggest minor revisions for clarity and increased significance.

We thank the editor and reviewers very much for their kindness and generosity with their time.

REVIEWER COMMENTS

Reviewer #1 (Remarks to the Author):

During the revision the authors removed the wrong statement about the impact of the low oxygen saturation water over the Chukchi Plateau on the large area of the Canada Basin, and now focus mainly on the phenomenon over the Chukchi Plateau itself. I find that the paper is convincing now.

I have one major comment on the explanation for the penetration of low oxygen saturation water from ESS shelf-slope to Chukchi Plateau. The authors attribute the penetration to the shrinkage of Beaufort Gyre and the Arctic Atlantification. This is also reflected in the paper title. The explanation is not fully correct. The shrinkage of the Beaufort Gyre is certainly one reason, but the Atlantification is not. The other main cause besides Beaufort Gyre shrinkage is the enhanced cyclonic circulation mode of the upper Arctic Ocean under positive Arctic Oscillation (possibly facilitated by cyclones intrusion as well). On average, the Arctic Oscillation was in a positive phase in the second half of the 2010s. Positive Arctic Oscillation releases surface freshwater (low salinity water) from the Eurasian Basin and Makarov Basin, thus enhancing the cyclonic circulation mode (Morison2021, Wang2022). This can enhance eastward penetration of halocline water and Atlantic Water, and the low oxygen saturation water originally over the ESS shelf-slope as well. Dynamically, it is the shrinkage of the anticyclonic Beaufort Gyre and the expansion of the cyclonic circulation together that caused the phenomenon described in this paper. The propagation of the signal of Atlantification toward the Canada Basin could also be strengthened by the stronger cyclonic circulation mode, but the Atlantification itself is not the dynamic cause of the penetration of low oxygen saturation water to the Chukchi Plateau. Certainly, both the shrinkage of the Beaufort Gyre and the enhancement of the cyclonic circulation mode are intensified by Arctic sea ice decline (Wang2022).

The shift of the boundary between Pacific derived and Atlantic derived waters towards the Mendeleev Ridge occurred in the 1990s following the strongly positive Arctic Oscillation (the strongest on average for the past six decades). Hydrography observations have shown this (Carmack1995, Morison1998, Steele1998). In a period with weak Beaufort Gyre, the penetration of the boundary to the Canada Basin under positive Arctic Oscillation is even stronger (Wang2021). It is worth mentioning the extreme event in the 1990s in the Discussion section.

It is required to polish the English writing.

Lit:

Morison, J., Kwok, R., Dickinson, S., Andersen, R., Peralta-Ferriz, C., Morison, D., et al. (2021). The Cyclonic Mode of Arctic Ocean Circulation. *J. Phys. Oceanog.* 51, 1053-1075. doi: 10.1175/JPO-D-20-0190.1

Wang Q and Danilov S (2022) A Synthesis of the Upper Arctic Ocean Circulation During 2000-2019: Understanding the Roles of Wind Forcing and Sea Ice Decline. *Front. Mar. Sci.* 9:863204. doi: 10.3389/fmars.2022.863204

Carmack, E. C., Macdonald, R., Perkin, R. G., McLaughlin, F. A., and Pearson, R. J. (1995). Evidence for Warming of Atlantic Water in the Southern Canadian Basin of the Arctic Ocean: Results from the Larsen-93 Expedition. *Geophys. Res. Lett.* 22, 1061–1064. doi: 10.1029/95GL00808

Morison, J., Steele, M., and Andersen, R. (1998). Hydrography of the Upper Arctic Ocean Measured from the Nuclear Submarine U.S.S. Pargo. *Deep-Sea. Res. I* 45, 15–38. doi: 10.1016/S0967-0637(97)00025-3

Steele, M., and Boyd, T. (1998). Retreat of the Cold Halocline Layer in the Arctic Ocean. *J. Geophys. Res. - Ocean.* 103, 10419–10435. doi: 10.1029/98JC00580

Wang Q., Danilov S., Sidorenko D., Wang X. (2021). Circulation Pathways and Exports of Arctic River Runoff Influenced by Atmospheric Circulation Regimes. *Front. Mar. Sci.* 8, 1153. doi: 10.3389/fmars.2021.707593

Some minor comments are given below.

Lines33-35. Sea ice decline could freshen some areas of the Arctic Ocean, but it could also increase upper ocean salinity in other parts of the Arctic Ocean due to its impact on ocean circulation (Wang2022). The current sentence neglects the second aspect and should be revised.

L36-37. The knowledge about PW inflow has been updated in Woodgate2021. Please update your sentence. Importantly, it is wrong to use two individual years to express trend. A trend estimate for the observation period was provided by Woodgate2021.

Lit:

R. A. Woodgate and C. Peralta-Ferriz, Warming and freshening of the Pacific inflow to the Arctic from 1990-2019 implying dramatic shoaling in Pacific Winter water ventilation of the Arctic water column, *Geophysical Research Letters*, vol. 48, e2021GL092528, 2021. doi:

<https://doi.org/10.1029/2021GL092528>

L44, with - by

L45, do you mean “salinification” of the halocline?

L45, do you mean “weakening of the ocean stratification” in the halocline?

L48, Wang2022 mentioned above well explains the different responses of the western and eastern Arctic to Arctic sea ice retreat.

L50, in a - in the

L51-56, the previous modelling study (Wang2021) details the response of the boundary between Pacific-Atlantic origin waters and the circulation of runoff water to different modes of Arctic atmospheric circulation, which explains observations mentioned in this paragraph.

L62, was - were

L63, - never been studied

L62-63, Was this appearance of unusual water described before? These sentences leave an impression that this phenomenon has been described in past studies. In this case, please add citations. If not the case, then please revise these sentences.

L97, “Pacific/Atlantic boundary region” is confusing. For example, it is clearer to say “... in the boundary region between Pacific and Atlantic derived waters”. In other places in the paper as well.

L98 on a - on the

L103, by - through

L119, at the depths or between the depth ranges of ?

L128, characterizing - resulting in

L131, remove “at”

L151, - only with this ...

L185-187, "The westward (eastward) flow of the southern (northern) branch of the gyre overshoot (returned to) the southern (northern) Chukchi Plateau." This sentence is hard to follow. Please revise.

L187, shrink - shrinkage

L188-189, "The extent of the Beaufort Gyre is governed by the strength of the Beaufort High, as described above." This sentence is not fully correct. The extent of the Beaufort Gyre corresponds to the spatial distribution of freshwater (low salinity), which is largely determined by the spatial pattern of wind curl. Note that "wind curl" is different from the strength of Beaufort High. Or, „strength of Beaufort High“ is not clear: the spatial scale of wind curl or the magnitude of wind curl. Also see comment below.

L189-193. This part of the text needs revision because the explanation is not fully correct.

The accumulation and release of freshwater in the Canada Basin is governed by wind-driven convergence and divergence anomalies, for which wind curl is a good indicator (Fig3B of Wang2022). The wind curl anomaly was not strongly negative before 2000 (Fig3A of Wang2022) so we do not expect a strong Beaufort Gyre for that period. The collapse of the Beaufort High caused a (southeast) shrinkage of the wind curl negative anomaly in the second half of the 2010s, which drives the shrinkage of the Beaufort Gyre.

You mentioned the impact of sea ice compactness in the past. Yes, sea ice can impact the ocean surface stress and thus ocean surface Ekman transport, but wind curl is a first order indicator for the evolution of the Beaufort Gyre. As shown by the blue line in Fig. 3A of Wang2022, we do not expect a stronger Beaufort Gyre averaged over the few decades before 2000. Therefore, in Fig. 1e, the forcing of the "BH" (the magnitude of wind curl) on the ocean should be weaker than in Fig. 1d. The difference in the BG between d and e is not only due to sea ice decline, but also due to the difference in wind curl.

In Fig. 1c, the recent shrinkage of the Beaufort Gyre and the penetration of LowOxy water onto CP is not only due to the collapse of BH, but also due to positive Arctic Oscillation, which strengthens the upper ocean cyclonic circulation (Morison2021, Wang2022). Both the impacts of Arctic Oscillation and shrinkage of BH on the ocean are intensified by sea ice decline (Wang2022).

L199, remove "was"

L207, L208, of salinity - with salinity

L209, the influences on what?

L212, characterizing - influencing

L216, - decline

L216-219, The Arctic Atlantification previously identified in the “eastern Eurasian Basin” can be largely explained by sea ice decline through its impacts on ocean surface stress and upper ocean circulation. See the explanation in Wang2022, and the argument therein that the contribution from salinity changes in the Barents Sea associated with sea ice decline is relatively small.

L223 “cyclonic atmospheric circulation”, here it would be better to also refer to the positive phase of the Arctic Oscillation, which enhanced cyclonic upper ocean circulation in the second half of the 2010s.

L223, - - change to “With the strengthening and expansion of the cyclonic circulation and the shrinkage of the anticyclonic circulation, the LHW and AW that carry signals of Arctic Atlantification could further flow ...”

L246-247, As commented above, it is rather the concurrent expansion of the cyclonic circulation and shrinkage of the Beaufort Gyre that determines the fate of the water originally located on the ESS shelf-slope.

L267, would not be - was not

L272, - increase along the northward flow primarily ...

L281-282, change to “... in the northeastern Canada Basin.”

L283, remove “be”

L287, LHW passes Lomonosov Ridge to enter the Makarov Basin. Why do you mention “Mendeleev Ridge” here?

L275-287, In Fig. 4g and 4h, the oxygen saturation in the interior of the Canada Basin is lower in g than in h. To what an extent the displayed difference in oxygen saturation is influenced by the choice of isopycnal layer for showing the plots? Could you show a vertical transect of oxygen saturation

through the Canada Basin for the two periods separately? The oxygen saturation in the interior of the Canada Basin is influenced not only by horizontal advection, but also by vertical motion.

L299, The shrinkage of the Beaufort Gyre took place over years. If you see that the western boundary of the Beaufort Gyre (northward flow) is located over the Chukchi Plateau in 2017, it means that the process of the shrinkage had started before 2017 already. Arctic Oscillation was positive in 2015, which can contribute to the release of freshwater from the Makarov Basin and Chukchi Plateau, thus the eastward shrinkage of the western boundary of the Beaufort Gyre.

L300, Fig5b shows some seasonality.

L316, Does the model reproduce the observed changes in the strength and spatial extent of the Beaufort Gyre shown in Fig.4?

L317-319, The concurrent shrinkage of the Beaufort Gyre and the strengthening of the cyclonic circulation facilitate the transport of the low oxygen saturation water from the ESS shelf-slope to the Chukchi Plateau. I do not see dynamic impacts of Atlantification on the transport of this water.

L333-338, check the grammar of this long sentence.

L360-361, - resulting in quicker and wider formation of low oxygen saturation water

L394 – scales

L406 “strong ventilation in the winter and higher alkalinity of AW also...” Check grammar.

Reviewer #3 (Remarks to the Author):

I believe this draft of the manuscript is appropriate for prompt publication in Nature Communications. The organization is clear; the findings are supported; sufficient context and discussion are provided to aid a wide readership in understanding the conclusions and implications; the reporting of previously unobserved biogeochemical anomalies in response to changing Arctic circulation are noteworthy and a plausible mechanism is identified and supported with observational and model analyses. The authors made defensible revisions in response to my prior review, including presenting new numerical model analyses in support of their hypothesis. I've provided several minor comments below for the authors' further consideration, if they choose.

Abs,L30. I suggest instead "This flow likely transports the low oxygen and acidified water towards the fishable area; similar biogeochemical properties had previously been observed only on the shelf and slope of the East Siberian Sea."

Int,L45. Perhaps "upwelling" instead of "uplifts."

Int, L63. I suggest deleting this sentence, and instead adding to the end of the previous sentence: "...boundary region for the first time." Or in the next paragraph, you could write "Here, we describe these biogeochemically anomalous conditions for the first time, and propose a mechanism driving these changes by combining data...".

Int,L66-68. I think it would be more accurate to say that describing the presence and drivers of these biogeochemical anomalies is a vital preliminary step--we can't assess ecosystem responses until we recognize the hydrographic changes underpinning them.

Res,L127. See also comment regarding S1,L38. Transmissometry could provide a snapshot of suspended particulate matter, but I don't think it provides enough evidence to rule out episodic particle flux events or other contributing factors to water column and sediment oxygen demand. So I personally think it would be more accurate to say that there is not evidence of high levels suspended sediments and particles that might enhance local oxygen uptake, unlike the conditions observed in the ESS. Thus you expect that biogeochemical anomalies transported from elsewhere are an important factor in the observed low oxygen conditions. That makes the same point without claiming that you can rule out local changes in oxygen demand, which would be hard to prove. Your discussion around Line 352 seems to relate to this assertion as well, so it might be worth bringing up earlier.

Res,L185-187. I find this sentence to be complicated to understand with all the parentheses. Would you please rewrite this as two sentences, one for each branch of the flow?

Fig. 5 and associated discussion, ~L316. It is also not unusual for regional ocean models to have biases in their vertical structure (poor characterization of onshore shoaling of isopycnal layers) due to either model and bathymetry resolution or issues with boundary forcings. You noted in particular that the Beaufort High collapsed in 2017--so my expectation is that the underlying pycnocline should have shoaled, as observed. If either that pressure or changing ocean boundary conditions with increased Atlantic water inputs were not accounted for in the model forcings, I imagine that could result in the similar isopycnal structure you see in the two model periods compared to the offset observed in observations. If you check and this is what happened, I suggest revising the text to acknowledge this as a model limitation that helps explain the discrepancy you noted on Line 316.

Dis,L337. "which [had previously] only ever appeared"

Overall I think the discussion of potential ecological impacts is more compelling in this draft. I have two suggestions to consider:

Dis,L374. Pacific cod and Greenland cod were recently suggested to be the same species based on genetic evidence. There are few Arctic observations attributed to either (<https://obis.org/taxon/254538>) but any general population connectivity would appear to span the Chukchi and Beaufort Seas. I believe the Pacific cod fishery is roughly an order of magnitude larger than either species you mention here, and consequently could be of greater commercial importance in a warming Arctic (just as you mention Atlantic cod might become). Thus oxygen loss that could "only" happen in this region could still be ecologically important to the broader Arctic and subarctic ecosystem.

Dis, Line 380. Environmental hypoxia is often designated as a threshold--and that value of ~60 $\mu\text{mol kg}^{-1}$ is heavily tilted towards warm water systems, particularly the Gulf of Mexico. Cold water fish tend to have higher hypoxia thresholds (<https://www.nature.com/articles/s41586-020-2721-y>), implying ecological consequences could occur at higher O_2 . Using the hypoxia traits in the supplemental table of this reference for Greenland cod, this species might be expected to experience hypoxia at 75%-85% of O_2sat (at 0C). In other words, the recent anomalously low O_2 might be expected to limit connectivity between the Greenland and Pacific populations (if they really are the same species). If you choose, you could use this or hypoxia thresholds for other coldwater species to demonstrate that current low O_2 levels already have the potential to cause ecological impacts compared to prior conditions.

S1,L38. I think that transmissometry does not say much about prior carbon export events or diffusive sediment oxygen fluxes. So I think the phrase "could not" expresses too much certainty.

S2,L60. Throughout, I suggest "gridded" instead of "grid" datasets. Also, in general I suggest "uncertainties" rather than "errors," because it sounds like each of these variances are driven by real environmental phenomenon rather than methodological problems.

Responses to comments of Reviewer #1 on “Beaufort Gyre shrinkage and Atlantification induced an anomalous biogeochemical event in the western Arctic Ocean” by S. Nishino, J. Jung, K.-H. Cho, W. J. Williams, A. Fujiwara, A. Murata, M. Itoh, E. Watanabe, M. Aoyama, M. Yamamoto-Kawai, T. Kikuchi, E. J. Yang, and S.-H. Kang (Paper # NCOMMS-22-30054A).

We would like to express our gratitude once again for valuable comments from the reviewer, which improved our paper significantly. We have revised the manuscript in line with suggestions from the reviewer. Our respective responses are shown below.

Reviewer #1 (Remarks to the Author):

During the revision the authors removed the wrong statement about the impact of the low oxygen saturation water over the Chukchi Plateau on the large area of the Canada Basin, and now focus mainly on the phenomenon over the Chukchi Plateau itself. I find that the paper is convincing now.

I have one major comment on the explanation for the penetration of low oxygen saturation water from ESS shelf- slope to Chukchi Plateau. The authors attribute the penetration to the shrinkage of Beaufort Gyre and the Arctic Atlantification. This is also reflected in the paper title. The explanation is not fully correct. The shrinkage of the Beaufort Gyre is certainly one reason, but the Atlantification is not. The other main cause besides Beaufort Gyre shrinkage is the enhanced cyclonic circulation mode of the upper Arctic Ocean under positive Arctic Oscillation (possibly facilitated by cyclones intrusion as well). On average, the Arctic Oscillation was in a positive phase in the second half of the 2010s. Positive Arctic Oscillation releases surface freshwater (low salinity water) from the Eurasian Basin and Makarov Basin, thus enhancing the cyclonic circulation mode (Morison2021, Wang2022). This can enhance eastward penetration of halocline water and Atlantic Water, and the low oxygen saturation water originally over the ESS shelf-slope as well. Dynamically, it is the shrinkage of the anticyclonic Beaufort Gyre and the expansion of the cyclonic circulation together that caused the phenomenon described in this paper. The propagation of the signal of Atlantification toward the Canada Basin could also be strengthened by the stronger cyclonic circulation mode, but the Atlantification itself is not the dynamic cause of the penetration of low oxygen saturation water to the Chukchi Plateau. Certainly, both the shrinkage of the Beaufort Gyre and the enhancement of the cyclonic circulation mode are intensified by Arctic sea ice decline (Wang2022).

Response:

We deleted the word “Atlantification” from the title. We largely modified the sections of Introduction and Results to include the viewpoint of Arctic Oscillation, a positive phase of

which can cause the enhanced cyclonic circulation mode of the upper Arctic Ocean. In the Discussion section, we mentioned shortly the Atlantification whose signal could be carried by the cyclonic circulation.

The shift of the boundary between Pacific derived and Atlantic derived waters towards the Mendeleev Ridge occurred in the 1990s following the strongly positive Arctic Oscillation (the strongest on average for the past six decades). Hydrography observations have shown this (Carmack1995, Morison1998, Steele1998). In a period with weak Beaufort Gyre, the penetration of the boundary to the Canada Basin under positive Arctic Oscillation is even stronger (Wang2021). It is worth mentioning the extreme event in the 1990s in the Discussion section.

Response:

In the Discussion section, we described the oxygen saturation distribution obtained from the 1994 Arctic Ocean Section (lines 370-381, see also Supplementary Fig. 7). This observation was conducted under a condition that the shift of the boundary between the Pacific and Atlantic waters toward the Mendeleev Ridge occurred in the 1990s following the strongly positive Arctic Oscillation.

It is required to polish the English writing.

Response:

We requested native speakers of English to proofread our English writing.

Some minor comments are given below.

Lines33-35. Sea ice decline could freshen some areas of the Arctic Ocean, but it could also increase upper ocean salinity in other parts of the Arctic Ocean due to its impact on ocean circulation (Wang2022). The current sentence neglects the second aspect and should be revised.

Response:

We revised the sentence as follows (see lines 33-35).

“As the sea ice decline, the response of the upper ocean circulation to winds can be enhanced and the freshwater in the eastern Arctic basins is more likely to shift toward the Beaufort Gyre region in the western Arctic (Wang and Danilov, 2022; Wang et al., 2019).”

L36-37. The knowledge about PW inflow has been updated in Woodgate2021. Please update your sentence. Importantly, it is wrong to use two individual years to express trend. A trend estimate for the observation period was provided by Woodgate2021.

Response:

We revised the sentence as follows (see lines 36-38).

“Bering Strait mooring observations from 1990 to 2019 show increasing northward flow of Pacific water (PW), spring/fall warming, and winter freshening (Woodgate and Peralta-Ferriz, 2021).”

L44, with - by

Response: Corrected (see line 46).

L45, do you mean “salinification” of the halocline?

L45, do you mean “weakening of the ocean stratification” in the halocline?

Response:

We revised the sentence as follows (see lines 46-49).

“Atlantification accompanies the salinification of halocline above the AW layer, weakening of the halocline stratification, shoaling of the AW layer with higher nutrients, and a possible increase in phytoplankton biomass by the vertical nutrient supply (Polyakov et al., 2020) and the phytoplankton transport by AW (Oziel et. Al., 2020).”

L48, Wang2022 mentioned above well explains the different responses of the western and eastern Arctic to Arctic sea ice retreat.

Response:

We added Wang and Danilov (2022) to a reference that studied the different responses of the western and eastern Arctic to Arctic sea ice retreat.

L50, in a - in the

Response:

Corrected. But we have changed the phrase “in the boundary region between the PW and AW influences” to “in the region toward the north of ESS and the western Chukchi Sea including the Chukchi Plateau where PW, AW, and ESSW complicatedly intersect (hereafter referred to as the intersectional water region)” to be more accurate. Please see lines 69-71.

L51-56, the previous modelling study (Wang2021) details the response of the boundary between Pacific-Atlantic origin waters and the circulation of runoff water to different modes of Arctic atmospheric circulation, which explains observations mentioned in this paragraph.

Response:

This paragraph was largely modified to start with the description of the study by Wang et al. (2021). Please see lines 52-71.

L62, was - were

Response: Corrected (see line 75).

L63, - never been studied

Response:

We have changed the phrase “never studied before” to “for the first time” to be more accurate. Please see lines 74-75.

L62-63, Was this appearance of unusual water described before? These sentences leave an impression that this phenomenon has been described in past studies. In this case, please add citations. If not the case, then please revise these sentences.

Response:

We revised the sentence as follows (see lines 74-75).

“During the survey, unusually low dissolved oxygen and acidified water were observed on the Chukchi Plateau, an ocean ridge north of the western Chukchi Sea, for the first time.”

L97, “Pacific/Atlantic boundary region” is confusing. For example, it is clearer to say “... in the boundary region between Pacific and Atlantic derived waters”. In other places in the paper as well.

Response:

Here, we simply changed the phrase “Pacific/Atlantic boundary region” to “Chukchi Plateau”. Elsewhere, as noted above, we have changed the phrase “the boundary region between the PW and AW influences” to “the intersectional water region”. Please see lines 69-71.

L98 on a - on the

Response: Corrected (see line 111).

L103, by - through

Response: Corrected (see lines 115 and 132).

L119, at the depths or between the depth ranges of ?

Response:

We have changed the phrase “at the depths of 100 and 300 m” to “between the depth range of 100 and 300 m”. Please see line 131.

L128, characterizing - resulting in

Response:

We have changed the phrase “the organic matter decomposition characterizing the anomalous water” to “high-level suspended sediments and particles that might promote local microbial oxygen consumption and CO₂ production”. Please see lines 137-138.

L131, remove “at”

Response: Corrected (see line 141).

L151, - only with this ...

Response: Corrected (see lines 159-160).

L185-187, “The westward (eastward) flow of the southern (northern) branch of the gyre overshot (returned to) the southern (northern) Chukchi Plateau.” This sentence is hard to follow. Please revise.

Response:

We revised the sentence as follows (see lines 207-208).

“In 2008–2016 (Fig. 4b), the westward flow of the southern part of the Beaufort Gyre overshot the southern Chukchi Plateau (CP), and the return flow passed across the northern CP.”

L187, shrink - shrinkage

Response: Corrected (see line 209).

L188-189, “The extent of the Beaufort Gyre is governed by the strength of the Beaufort High, as described above.” This sentence is not fully correct. The extent of the Beaufort Gyre corresponds to the spatial distribution of freshwater (low salinity), which is largely determined by the spatial pattern of wind curl. Note that “wind curl” is different from the

strength of Beaufort High. Or, „strength of Beaufort High“ is not clear: the spatial scale of wind curl or the magnitude of wind curl. Also see comment below.

Response:

We revised the sentence as follows (see lines 177-178).

“The Beaufort High induces a negative curl of winds that converges the surface freshwater within the Beaufort Gyre (BG), and the spatial distribution of freshwater determines the extent of the BG.”

L189-193. This part of the text needs revision because the explanation is not fully correct. The accumulation and release of freshwater in the Canada Basin is governed by wind-driven convergence and divergence anomalies, for which wind curl is a good indicator (Fig3B of Wang2022). The wind curl anomaly was not strongly negative before 2000 (Fig3A of Wang2022) so we do not expect a strong Beaufort Gyre for that period. The collapse of the Beaufort High caused a (southeast) shrinkage of the wind curl negative anomaly in the second half of the 2010s, which drives the shrinkage of the Beaufort Gyre.

Response:

We revised the sentences as follows (see lines 213-217).

“In the 1950s–1980s (Fig. 4c), the Beaufort Gyre circulation was weak because the wind curl over the Canada Basin was not strongly negative in this period (Wang, and Danilov, 2022). A heavy and less mobile sea ice condition in this period might also inhibit the input of wind curl to the ocean (Yang, 2009). Because the Arctic Oscillation was a negative phase in this period, the cyclonic ocean circulation would not expand to the study area.”

Regarding the collapse of the Beaufort High, we mentioned this point in the Discussion section. Please see lines 364-369.

You mentioned the impact of sea ice compactness in the past. Yes, sea ice can impact the ocean surface stress and thus ocean surface Ekman transport, but wind curl is a first order indicator for the evolution of the Beaufort Gyre. As shown by the blue line in Fig. 3A of Wang2022, we do not expect a stronger Beaufort Gyre averaged over the few decades before 2000. Therefore, in Fig. 1e, the forcing of the “BH” (the magnitude of wind curl) on the ocean should be weaker than in Fig. 1d. The difference in the BG between d and e is not only due to sea ice decline, but also due to the difference in wind curl.

Response:

We revised Fig. 1e so that the forcing of the Beaufort High (the magnitude of wind curl) on the ocean was weaker than in Fig. 1d. Also, an arrow representing the status of the Arctic Oscillation was added to the top of Fig. 1c-e.

In Fig. 1c, the recent shrinkage of the Beaufort Gyre and the penetration of LowOxy water onto CP is not only due to the collapse of BH, but also due to positive Arctic Oscillation, which strengthens the upper ocean cyclonic circulation (Morison2021, Wang2022). Both the impacts of Arctic Oscillation and shrinkage of BH on the ocean are intensified by sea ice decline (Wang2022).

Response:

As mentioned above, an arrow representing the status of the Arctic Oscillation (AO) was added to the top of Fig. 1c-e. In Fig. 1c, the arrow indicates a positive AO.

In the first paragraph of the Discussion section, we described that the Arctic sea ice decline intensified both the Beaufort Gyre and cyclonic ocean circulation (Wang and Danilov, 2022).

L199, remove “was”

Response: Corrected (see line 221).

L207, L208, of salinity - with salinity

Response: Corrected (see lines 237 and 239).

L209, the influences on what?

Response:

We used the term “intrusions” instead of “influences” (see line 240).

L212, characterizing - influencing

Response:

The sentence including this word was deleted because it contained a description of the Atlantification.

L216, - decline

Response:

The sentence including this word was deleted because it contained a description of the Atlantification.

L216-219, The Arctic Atlantification previously identified in the “eastern Eurasian Basin”

can be largely explained by sea ice decline through its impacts on ocean surface stress and upper ocean circulation. See the explanation in Wang2022, and the argument therein that the contribution from salinity changes in the Barents Sea associated with sea ice decline is relatively small.

Response:

In association with the sea ice decline, the shift of freshwater from the cyclonic ocean circulation region to the Beaufort Gyre region develops Atlantification in the eastern Arctic (Wang and Danilov, 2022). This point was described in lines 358-360 in the Discussion section. And the description on the Barents Sea was removed.

L223 “cyclonic atmospheric circulation“, here it would be better to also refer to the positive phase of the Arctic Oscillation, which enhanced cyclonic upper ocean circulation in the second half of the 2010s.

Response:

We revised the sentence as follows (see lines 229-231).

“The enlarged LHW and AW intrusions were presumably associated with the intensification of AW supply induced by the sea ice decline (Atlantification; Wang et al., 2020) and the positive phase of the Arctic Oscillation, which enhanced the cyclonic ocean circulation, in the second half of the 2010s.”

L223, - - change to “With the strengthening and expansion of the cyclonic circulation and the shrinkage of the anticyclonic circulation, the LHW and AW that carry signals of Arctic Atlantification could further flow ...”

Response:

We revised the sentence as follows (see lines 231-234).

“With the strengthening and expansion of the cyclonic ocean circulation and the shrinkage of the anticyclonic Beaufort Gyre circulation, the LHW and AW could further flow ...”

L246-247, As commented above, it is rather the concurrent expansion of the cyclonic circulation and shrinkage of the Beaufort Gyre that determines the fate of the water originally located on the ESS shelf-slope.

Response:

We revised the sentence as follows (see lines 256-257).

“The concurrent expansion of the cyclonic ocean circulation and shrinkage of the Beaufort Gyre would determine the fate of the water originally located on the ESS shelf-slope.”

L267, would not be - was not

Response: Corrected (see line 276).

L272, - increase along the northward flow primarily ...

Response: Corrected (see line 281).

L281-282, change to "... in the northeastern Canada Basin."

Response: Corrected (see line 289).

L283, remove "be"

Response: Corrected (see line 292).

L287, LHW passes Lomonosov Ridge to enter the Makarov Basin. Why do you mention "Mendelev Ridge" here?

Response:

We revised the sentence as follows not to mention the Mendelev Ridge (see lines 300-302).

"This shift would result from the penetration of LHW with a high oxygen saturation to the Makarov Basin by the expanded cyclonic ocean circulation in 2017–2020 under the positive Arctic Oscillation."

L275-287, In Fig. 4g and 4h, the oxygen saturation in the interior of the Canada Basin is lower in g than in h. To what an extent the displayed difference in oxygen saturation is influenced by the choice of isopycnal layer for showing the plots? Could you show a vertical transect of oxygen saturation through the Canada Basin for the two periods separately? The oxygen saturation in the interior of the Canada Basin is influenced not only by horizontal advection, but also by vertical motion.

Response:

The lower oxygen saturation levels in the Canada Basin in 2017–2020 than in 2008–2016 were observed in the whole LHW (salinity = 34–34.5) layer (see lines 293-295). We added vertical sections of oxygen saturation in 2017–2020 and 2008–2016 as Supplementary Fig. 5a, b. We also added the description on the vertical mixing of oxygen saturation using a conservative tracer, NO (Supplementary Fig. 5c, d). Please see lines 295-299 and Supplementary Discussion 3.

L299, The shrinkage of the Beaufort Gyre took place over years. If you see that the western

boundary of the Beaufort Gyre (northward flow) is located over the Chukchi Plateau in 2017, it means that the process of the shrinkage had started before 2017 already. Arctic Oscillation was positive in 2015, which can contribute to the release of freshwater from the Makarov Basin and Chukchi Plateau, thus the eastward shrinkage of the western boundary of the Beaufort Gyre.

Response:

We added the description of delayed oceanic response to surface forcing as follows (see lines 208-213).

“The shrinkage of the Beaufort Gyre (BG) in 2017–2020 would be a response of ocean circulation to the change in the Arctic Oscillation (AO) phase in the mid-2010s. The time scale of the delayed oceanic response to surface forcing (a few years) is consistent with the previous studies (Yoshizawa et al., 2015). That is, a positive AO in the second half of the 2010s caused a southeast shrinkage of the negative anomaly area of wind curl over the Canada Basin (Wang and Danilov, 2022), which drove the shrinkage of the BG with a delay of several years from the change in the AO phase.”

Please also see lines 312-315 as follows.

“Associated with this transport, the appearance of low oxygen saturation water passing through the Chukchi Plateau has become more pronounced since 2017 (Fig. 5b), which is consistent with the year when the observed Beaufort Gyre shrunk in 2017 after a several-year delay from the change in the Arctic Oscillation phase in the mid-2010s.”

L300, Fig5b shows some seasonality.

Response:

We revised the sentence as follows (see lines 318-320).

“The tracer concentration of the water seems to represent a seasonality with its increase in winter when winds are generally strong (Martin et al., 2014; Stopa et al., 2016).”

L316, Does the model reproduce the observed changes in the strength and spatial extent of the Beaufort Gyre shown in Fig.4?

Response:

We added the simulated sea surface height as Supplementary Fig. 6 and described as follows (see lines 315-316).

“Our numerical model also indicates the shrinking of the Beaufort Gyre in 2017–2020 (Supplementary Fig. 6a) compared to 2013–2016 (Supplementary Fig. 6b).”

L317-319, The concurrent shrinkage of the Beaufort Gyre and the strengthening of the cyclonic circulation facilitate the transport of the low oxygen saturation water from the ESS

shelf-slope to the Chukchi Plateau. I do not see dynamic impacts of Atlantification on the transport of this water.

Response:

We deleted this sentence concerning the Atlantification. Your proposed sentence above was added to the first sentence of the Discussion section (see lines 355-357).

L333-338, check the grammar of this long sentence.

Response:

This sentence was deleted because it contained an incorrect explanation as you pointed out earlier in this review.

L360-361, - resulting in quicker and wider formation of low oxygen saturation water

Response: Corrected (see lines 403-404).

L394 – scales

Response: Corrected (see line 442).

L406 “strong ventilation in the winter and higher alkalinity of AW also...” Check grammar.

Response:

We revised the sentence as follows based on the native speakers’ proofread (see lines 453-454).

“strong winter ventilation and AW’s high alkalinity contribute to the sink of atmospheric CO₂ in the AW gateway area.”

We thank the editor and reviewers very much for their kindness and generosity with their time.

Responses to comments of Reviewer #3 on “Beaufort Gyre shrinkage and Atlantification induced an anomalous biogeochemical event in the western Arctic Ocean” by S. Nishino, J. Jung, K.-H. Cho, W. J. Williams, A. Fujiwara, A. Murata, M. Itoh, E. Watanabe, M. Aoyama, M. Yamamoto-Kawai, T. Kikuchi, E. J. Yang, and S.-H. Kang (Paper # NCOMMS-22-30054A).

Again, we deeply appreciate valuable comments of the reviewer, which improved our paper significantly. We have revised the manuscript in line with suggestions from the reviewer. Our respective responses are shown below.

Reviewer #3 (Remarks to the Author):

I believe this draft of the manuscript is appropriate for prompt publication in Nature Communications. The organization is clear; the findings are supported; sufficient context and discussion are provided to aid a wide readership in understanding the conclusions and implications; the reporting of previously unobserved biogeochemical anomalies in response to changing Arctic circulation are noteworthy and a plausible mechanism is identified and supported with observational and model analyses. The authors made defensible revisions in response to my prior review, including presenting new numerical model analyses in support of their hypothesis. I've provided several minor comments below for the authors' further consideration, if they choose.

Abs,L30. I suggest instead "This flow likely transports the low oxygen and acidified water towards the fishable area; similar biogeochemical properties had previously been observed only on the shelf and slope of the East Siberian Sea."

Response:

We revised the sentence as follows (see lines 29-31).

“This flow likely transports the low oxygen and acidified water toward the fishable area; similar biogeochemical properties had previously been observed only on the shelf-slope north of the East Siberian Sea.”

Int,L45. Perhaps "upwelling" instead of "uplifts."

Response:

The uplifts of nutrient-rich water are associated with shoaling of the AW layer. Thus, we revised the sentence as follows (see line 48).

“shoaling of the AW layer with higher nutrients”

Int, L63. I suggest deleting this sentence, and instead adding to the end of the previous sentence: "...boundary region for the first time." Or in the next paragraph, you could write "Here, we describe these biogeochemically anomalous conditions for the first time, and propose a mechanism driving these changes by combining data..."

Response:

We deleted the sentence and revised the rest of the text as follows (see lines 74-77).
 "During the survey, unusually low dissolved oxygen and acidified water were observed on the Chukchi Plateau, an ocean ridge north of the western Chukchi Sea, for the first time. Herein, we describe these biogeochemically abnormal conditions and propose a mechanism driving these changes (Fig. 1c–e) by combining data..."

Int,L66-68. I think it would be more accurate to say that describing the presence and drivers of these biogeochemical anomalies is a vital preliminary step--we can't assess ecosystem responses until we recognize the hydrographic changes underpinning them.

Response:

We revised the sentence as follows (see lines 78-81).
 "Describing the presence and drivers of these biogeochemical anomalies is a vital preliminary step for Arctic ecosystem assessments (refs. 21,22); we can only assess ecosystem responses once we recognize the hydrographic changes underpinning them."

Res,L127. See also comment regarding S1,L38. Transmissometry could provide a snapshot of suspended particulate matter, but I don't think it provides enough evidence to rule out episodic particle flux events or other contributing factors to water column and sediment oxygen demand. So I personally think it would be more accurate to say that there is not evidence of high levels suspended sediments and particles that might enhance local oxygen uptake, unlike the conditions observed in the ESS. Thus you expect that biogeochemical anomalies transported from elsewhere are an important factor in the observed low oxygen conditions. That makes the same point without claiming that you can rule out local changes in oxygen demand, which would be hard to prove. Your discussion around Line 352 seems to relate to this assertion as well, so it might be worth bringing up earlier.

Response:

We deleted the description of transmissometry and its figure, and revised the rest of the text as follows (see lines 137-139). Here, we brought up the discussion on the organic matter deposition with references.
 "there is no evidence of high-level suspended sediments and particles that might promote local microbial oxygen consumption and CO₂ production, unlike the conditions observed in the ESS where much of terrestrial/marine organic matter is deposited on the bottom (refs. 24,25)."

Res,L185-187. I find this sentence to be complicated to understand with all the parentheticals. Would you please rewrite this as two sentences, one for each branch of the flow?

Response:

We revised the sentence as follows (see lines 207-208).

“In 2008–2016 (Fig. 4b), the westward flow of the southern part of the Beaufort Gyre overshot the southern Chukchi Plateau (CP), and the return flow passed across the northern CP.”

Fig. 5 and associated discussion, ~L316. It is also not unusual for regional ocean models to have biases in their vertical structure (poor characterization of onshore shoaling of isopycnal layers) due to either model and bathymetry resolution or issues with boundary forcings. You noted in particular that the Beaufort High collapsed in 2017--so my expectation is that the underlying pycnocline should have shoaled, as observed. If either that pressure or changing ocean boundary conditions with increased Atlantic water inputs were not accounted for in the model forcings, I imagine that could result in the similar isopycnal structure you see in the two model periods compared to the offset observed in observations. If you check and this is what happened, I suggest revising the text to acknowledge this as a model limitation that helps explain the discrepancy you noted on Line 316.

Response:

We added the description about conceivable reasons why the LHW penetration is not well reproduced in the model (see lines 335-339).

“The simulated penetration of LHW from the eastern Arctic might be underestimated because of the closed lateral boundary condition in the North Atlantic and insufficient vertical resolution. In addition, the lack of interannual variability and multi-decadal trends in riverine freshwater discharges might produce an unexpected model bias of stratification in the eastern Arctic (isohaline-surface structure of the LHW).”

Dis,L337. "which [had previously] only ever appeared"

Response: Corrected (see lines 356-357).

Overall I think the discussion of potential ecological impacts is more compelling in this draft. I have two suggestions to consider:

Dis,L374. Pacific cod and Greenland cod were recently suggested to be the same species based on genetic evidence. There are few Arctic observations attributed to either (<https://obis.org/taxon/254538>) but any general population connectivity would appear to

span the Chukchi and Beaufort Seas. I believe the Pacific cod fishery is roughly an order of magnitude larger than either species you mention here, and consequently could be of greater commercial importance in a warming Arctic (just as you mention Atlantic cod might become). Thus oxygen loss that could "only" happen in this region could still be ecologically important to the broader Arctic and subarctic ecosystem.

Response:

As you pointed out, Pacific cod and Greenland cod are the same species (Carr et al., 1999, doi:10.1139/cjz-77-1-19), but they are generally considered to be sedentary and nonmigratory species (Mikhail and Welch, 1989, doi:10.1007/BF00002475). In the Chukchi Sea, Pacific cod is expected to expand north and eastward if warming conditions continue, based on observational (ref. 60) and model (ref. 61) studies. However, such expansion might be inhibited by the anomalously low oxygen saturation water (less than the hypoxia threshold for the Pacific cod), which has already appeared on the Chukchi Plateau and is assumed to be enlarged in the future. This point is described in lines 428-431.

Dis, Line 380. Environmental hypoxia is often designated as a threshold--and that value of ~60 $\mu\text{mol kg}^{-1}$ is heavily tilted towards warm water systems, particularly the Gulf of Mexico. Cold water fish tend to have higher hypoxia thresholds (<https://www.nature.com/articles/s41586-020-2721-y>), implying ecological consequences could occur at higher O₂. Using the hypoxia traits in the supplemental table of this reference for Greenland cod, this species might be expected to experience hypoxia at 75%-85% of O₂sat (at 0C). In other words, the recent anomalously low O₂ might be expected to limit connectivity between the Greenland and Pacific populations (if they really are the same species). If you choose, you could use this or hypoxia thresholds for other coldwater species to demonstrate that current low O₂ levels already have the potential to cause ecological impacts compared to prior conditions.

Response:

Regarding the hypoxia threshold for cold-water fishes, we added descriptions, including the abovementioned sentences, as follows (lines 425-431).

“Cold-water fishes tend to have higher hypoxia thresholds (ref. 58). The Greenland cod, which distributes from Alaska to the western coast of Greenland, has a hypoxia threshold of ~40 % in oxygen saturation at 1 °C, and it would be higher oxygen saturation levels with ocean warming (ref. 59). The same species in the Chukchi Sea, i.e., the Pacific cod, is expected to expand north and eastward if warming conditions continue, based on observational (ref. 60) and model (ref. 61) studies. However, such expansion might be inhibited by the anomalously low oxygen saturation water, which has already appeared on the Chukchi Plateau and is assumed to be enlarged in the future.”

S1,L38. I think that transmissometry does not say much about prior carbon export events or

diffusive sediment oxygen fluxes. So I think the phrase "could not" expresses too much certainty.

Response:

We deleted the description of transmissometry and its figure as mentioned above. Here, we describe an assumption that the low oxygen saturation water is delivered from a remote region based on a conservative tracer, NO, as follows (see lines 38-42 in the Supplementary Information (SI)).

“The low oxygen saturation corresponded with low NO (Supplementary Fig. 1b), where NO is defined as $9[\text{NO}_3^-] + [\text{O}_2]$ ($\mu\text{mol kg}^{-1}$) and used as a quasi-conservative tracer that is independent of biological processes (ref. S1). Thus, the low oxygen saturation and low NO water are assumed to be delivered from a remote region rather than formed by local microbial oxygen consumption.”

S2,L60. Throughout, I suggest "gridded" instead of "grid" datasets. Also, in general I suggest "uncertainties" rather than "errors," because it sounds like each of these variances are driven by real environmental phenomenon rather than methodological problems.

Response:

We changed the term “grid” to “gridded” throughout the main text and SI. Also, we changed the term “error” to “uncertainty”.

We thank the editor and reviewers very much for their kindness and generosity with their time.

REVIEWERS' COMMENTS

Reviewer #1 (Remarks to the Author):

Thank the authors for their further effort to improve the manuscript. The dynamic processes are well explained now. I have only a couple of very minor comments:

Line 209-213

The shrinkage of the BG in 2017-2020 is associated with the collapse of the BH, that is, the area of the strong negative wind curl was confined to the southwest Canada Basin in this period.

Line 380-381

Shrinkage of the BG with strong northward flow along the CP is needed to transport the low oxygen saturation water toward the CP.

“strengthening the BG” could also mean enlargement (westward extension) of the BG, which will not lead to the transport toward the CP.

The authors need to proofread the text carefully in the next step, or the journal may check the language before publication? In some places the writing could be improved. Here just a few examples:

L33 decline – declines

L59 Arctic Oscillation (AO) was in a positive phase ...

L75 “western” Chukchi Sea should be “eastern” Chukchi Sea?

L80 ... we can assess ecosystem response only when we know hydrographic changes ...

L120, better add a “water” before “or”

L192 ... the AO was largely in a negative phase

L364-367 “2017” is within the period of “after the mid-2010s”. Rewrite these two sentences to make the flow logic.

L393-394, I cannot understand this sentence.

Responses to comments of Reviewer #1 on “Atlantic-origin water extension into the Pacific Arctic induced an anomalous biogeochemical event” by S. Nishino, J. Jung, K.-H. Cho, W. J. Williams, A. Fujiwara, A. Murata, M. Itoh, E. Watanabe, M. Aoyama, M. Yamamoto-Kawai, T. Kikuchi, E. J. Yang, and S.-H. Kang (Paper # NCOMMS-22-30054B).

We appreciate the time and effort that the editor and reviewer dedicated to providing feedback on our manuscript. We have incorporated most of the suggestions made by the reviewer. Our respective responses are shown below.

Reviewer #1 (Remarks to the Author):

Thank the authors for their further effort to improve the manuscript. The dynamic processes are well explained now. I have only a couple of very minor comments:

Line 209-213

The shrinkage of the BG in 2017-2020 is associated with the collapse of the BH, that is, the area of the strong negative wind curl was confined to the southwest Canada Basin in this period.

Response:

We revised the sentences as follows (see lines 213-215).

“The shrinkage of the BG in 2017–2020 is likely associated with the collapse of the BH in 2017. In this period, the area of the strong negative wind curl was confined to the southeastern CB.”

Line 380-381

Shrinkage of the BG with strong northward flow along the CP is needed to transport the low oxygen saturation water toward the CP.

“strengthening the BG” could also mean enlargement (westward extension) of the BG, which will not lead to the transport toward the CP.

Response:

We corrected the text as the reviewer indicated (see lines 382-384).

The authors need to proofread the text carefully in the next step, or the journal may check the language before publication?

Response:

We requested native speakers of English to proofread our English writing.

In some places the writing could be improved. Here just a few examples:

L33 decline – declines

Response: Corrected (see line 33).

L59 Arctic Oscillation (AO) was in a positive phase ...

Response: Corrected (see lines 58-59).

L75 “western” Chukchi Sea should be “eastern” Chukchi Sea?

Response: As the reviewer pointed out, “western Chukchi Sea” and “eastern Chukchi Sea” are ambiguous and confusing, so we simply termed “Chukchi Sea” (see line 70).

L80 ... we can assess ecosystem response only when we know hydrographic changes ...

Response: Corrected (see line 80).

L120, better add a “water” before “or”

Response: Corrected (see line 126).

L192 ... the AO was largely in a negative phase

Response: Corrected (see line 197).

L364-367 “2017” is within the period of “after the mid-2010s”. Rewrite these two sentences to make the flow logic.

Response: We changed “after the mid-2010s” to “after 2017” (see line 369).

L393-394, I cannot understand this sentence.

Response: We rewrote the sentence as follows (see lines 396-398).

“In this period, the water with a higher oxygen saturation would be transported to the CP, and therefore, the lowest oxygen saturation water might no longer be observed on the CP.”